# Minimax Optimal Alternating Minimization for Kernel Nonparametric Tensor Learning

**Taiji Suzuki**[*], **Heishiro Kanagawa**[†]
[*,†]Department of Mathematical and Computing Science, Tokyo Institute of Technology
[*]PRESTO, Japan Science and Technology Agency
[*]Center for Advanced Integrated Intelligence Research, RIKEN
s-taiji@is.titech.ac.jp, kanagawa.h.ab@m.titech.ac.jp

**Hayato Kobayash, Nobuyuki Shimizu, Yukihiro Tagami**
Yahoo Japan Corporation
{ hakobaya, nobushim, yutagami } @yahoo-corp.jp

## Abstract

We investigate the statistical performance and computational efficiency of the alternating minimization procedure for nonparametric tensor learning. Tensor modeling has been widely used for capturing the higher order relations between multimodal data sources. In addition to a linear model, a nonlinear tensor model has been received much attention recently because of its high flexibility. We consider an alternating minimization procedure for a general nonlinear model where the true function consists of components in a reproducing kernel Hilbert space (RKHS). In this paper, we show that the alternating minimization method achieves linear convergence as an optimization algorithm and that the generalization error of the resultant estimator yields the minimax optimality. We apply our algorithm to some multitask learning problems and show that the method actually shows favorable performances.

## 1 Introduction

Tensor modeling is widely used for capturing the higher order relations between several data sources. For example, it has been applied to spatiotemporal data analysis [19], multitask learning [20, 2, 14] and collaborative filtering [15]. The success of tensor modeling is usually based on the low-rank property of the target parameter. As in the matrix, the low-rank decomposition of tensors, e.g., *canonical polyadic (CP) decomposition* [10, 11] and *Tucker decomposition* [31], reduces the effective dimension of the statistical model, improves the generalization error, and gives a better understanding of the model based on an condensed representation of the target system.

Among several tensor models, linear models have been extensively studied from both theoretical and practical points of views [16]. A difficulty of the tensor model analysis is that typical tensor analysis problems usually fall under a non-convex problem and it is difficult to solve the problem. To overcome the computational difficulty, several authors have proposed convex relaxation methods [18, 23, 9, 30, 29]. Unfortunately, however, convex relaxation methods lose statistical optimality in favor of computational efficiency [28].

Another promising approach is the alternating minimization procedure which alternately updates each component of the tensor with the other fixed components. The method has shown a nice performance in practice. Moreover, its theoretical analysis has also been given by several authors [1, 13, 6, 3, 21, 36, 27, 37]. These theoretical analyses indicate that the estimator given by the

alternating minimization procedure has a good generalization error, with a mild dependency on the size of the tensor if the initial solution is properly set. In addition to the alternating minimization procedure, it has been shown that the Bayes estimator achieves the minimax optimality under quite weak assumptions [28].

Nonparametric models have also been proposed for capturing nonlinear relations [35, 24, 22]. In particular, [24] extended the linear tensor learning to the nonparametric learning problem using a kernel method and proposed a convex regularization method and an alternating minimization method. Recently, [14, 12] showed that the Bayesian approach has good theoretical properties for the nonparametric problem. In particular, it achieves the minimax optimality under weak assumptions. However, from a practical point of view, the Bayesian approach is computationally expensive compared with the alternating minimization approach. An interesting observation is that the practical performance of the alternating minimization procedure is quite good [24] and is comparable to the Bayesian one [14], although its computational efficiency is much better than that of the Bayesian one. Despite the practical usefulness of the alternating minimization, its statistical properties have not been investigated yet in the general nonparametric model.

In this paper, we theoretically analyze the alternating minimization procedure in the nonparametric model. We investigate its computational efficiency and analyze its statistical performance. It is shown that, if the true function is included in a reproducing kernel Hilbert space (RKHS), then the algorithm converges to an (a possibly local) optimal solution in *linear rate*, and the generalization error of the estimator achieves the *minimax optimality* if the initial point of the algorithm is in the $O(1)$ distance from the true function. Roughly speaking, the theoretical analysis shows that

$$\|\widehat{f}^{(t)} - f^*\|_{L_2}^2 = O_p\big(dKn^{-\frac{1}{1+s}}\log(dK) + dK\,(3/4)^t\,\big)$$

where $\widehat{f}^{(t)}$ is the estimated nonlinear tensor at the $t$th iteration of the alternating minimization procedure, $n$ is the sample size, $d$ is the rank of the true tensor, $K$ is the number of modes, and $s$ is the complexity of the RKHS. This indicates that the alternating minimization procedure can produce a minimax optimal estimator after $O(\log(n))$ iterations.

## 2 Problem setting: nonlinear tensor model

Here, we describe the model to be analyzed. Suppose that we are given $n$ input-output pairs $\{(x_i, y_i)\}_{i=1}^n$ that are generated from the following system. The input $x_i$ is a concatenation of $K$ variables, i.e., $x_i = (x_i^{(1)}, \cdots, x_i^{(K)}) \in \mathcal{X}_1 \times \cdots \times \mathcal{X}_K = \mathcal{X}$, where each $x_i^{(k)}$ is an element of a set $\mathcal{X}_k$ and is generated from a distribution $P_k$. We consider the regression problem where the outputs $\{y_i\}_{i=1}^n$ are observed according to the nonparametric tensor model [24]:

$$y_i = \sum_{r=1}^d \prod_{k=1}^K f_{(r,k)}^*(x_i^{(k)}) + \epsilon_i, \tag{1}$$

where $\{\epsilon_i\}_{i=1}^n$ represents an i.i.d. zero-mean noise and each $f_{(r,k)}^*$ is a component of the true function included in some RKHS $\mathcal{H}_{r,k}$. In this regression problem, our objective is to estimate the true function $f^*(x) = f^*(x^{(1)}, \ldots, x^{(K)}) = \sum_{r=1}^d \prod_{k=1}^K f_{(r,k)}^*(x^{(k)})$ based on the observations $\{(x_i, y_i)\}_{i=1}^n$. This model has been applied to several problems such as multitask learning, recommendation system and spatiotemporal data analysis. Although we focus on the squared loss regression problem, the discussion in this paper can be easily generalized to Lipschitz continuous and strongly convex losses as in [4].

**Example 1: multitask learning** Suppose that we have several tasks indexed by a two-dimensional index $(s, t) \in [M_1] \times [M_2]^1$, and each task $(s, t)$ is a regression problem for which there is a true function $g_{[s,t]}^*(x)$ that takes an input feature $w \in \mathcal{X}_3$. The $i$th input sample is given as $x_i = (s_i, t_i, w_i)$, which is a combination of task index $(s_i, t_i)$ and input feature $w_i$. By assuming that the true function $g_{[s,t]}^*$ is a linear combination of a few latent factors $h_r$ as

$$g_{[s,t]}^*(x) = \sum_{r=1}^d \alpha_{s,r}\beta_{t,r}h_r(w) \quad (x = (s, t, w)), \tag{2}$$

---

1 We denote by $[k] = \{1, \ldots, k\}$.

---

**Algorithm 1** Alternating minimization procedure for nonlinear tensor estimation

---

**Require:** Training data $D_n = \{(x_i, y_i)\}_{i=1}^n$, the regularization parameter $\lambda^{(n)}$, iteration number $T$.
**Ensure:** $\widehat{f} = \sum_{r=1}^d \hat{v}_r^{(T)} \prod_{k=1}^K \widehat{f}_{(r,k)}^{(T)}$ as the estimator
   **for** $t = 1, \ldots, T$ **do**
      Set $\tilde{f}_{(r,k)} = \widehat{f}_{(r,k)}^{(t-1)}$ $(\forall (r,k))$, and $\tilde{v}_r = \hat{v}_r^{(t-1)}$ $(\forall r)$.
      **for** $(r,k) \in \{1, \ldots, d\} \times \{1, \ldots, K\}$ **do**
         The $(r,k)$-element of $\tilde{f}$ is updated as

$$\tilde{f}'_{(r,k)} = \operatorname*{argmin}_{f_{(r,k)} \in \mathcal{H}_{r,k}} \left\{ \frac{1}{n} \sum_{i=1}^n \left[ y_i - \Big( f_{(r,k)} \prod_{k' \neq k} \tilde{f}_{(r,k')} + \sum_{r' \neq r} \tilde{v}_{r'} \prod_{k'=1}^K \tilde{f}_{(r',k')} \Big)(x_i) \right]^2 + C_n \|f\|_{\mathcal{H}_{r,k}}^2 \right\}. \quad (4)$$

$$\tilde{v}_r \leftarrow \|\tilde{f}'_{(r,k)}\|_n, \ \tilde{f}_{(r,k)} \leftarrow \tilde{f}'_{(r,k)}/\tilde{v}_r.$$
      **end for**
      Set $\widehat{f}_{(r,k)}^{(t)} = \tilde{f}_{(r,k)}$ $(\forall (r,k))$ and $\hat{v}_r^{(t)} = \tilde{v}_r$ $(\forall r)$.
   **end for**

---

and the output is given as $y_i = g^*_{[s_i, t_i]}(x_i) + \epsilon_i$ [20, 2, 14], then we can reduce the multitask learning problem for estimating $\{g^*_{[s,t]}\}_{s,t}$ to the tensor estimation problem, where $f_{(r,1)}(s) = \alpha_{s,r}$, $f_{(r,2)}(t) = \beta_{t,r}$, $f_{(r,3)}(w) = h_r(w)$.

## 3 Alternating regularized least squares algorithm

To learn the nonlinear tensor factorization model (1), we propose to optimize the regularized empirical risk in an alternating way. That is, we optimize each component $f_{(r,k)}$ with the other fixed components $\{f_{(r',k')}\}_{(r',k') \neq (r,k)}$. Basically, we want to execute the following optimization problem:

$$\min_{\{f_{(r,k)}\}_{(r,k)}: f_{(r,k)} \in \mathcal{H}_{r,k}} \ \frac{1}{n} \sum_{i=1}^n \left( y_i - \sum_{r=1}^d \prod_{k=1}^K f_{(r,k)}(x_i^{(k)}) \right)^2 + C_n \sum_{r=1}^d \sum_{k=1}^d \|f_{(r,k)}\|_{\mathcal{H}_{r,k}}^2, \quad (3)$$

where the first term is the loss function for measuring how our guess $\sum_{r=1}^d \prod_{k=1}^K f_{(r,k)}$ fits the data and the second term is a regularization term for controlling the complexity of the learning function. However, this optimization problem is not convex and is difficult to exactly solve.

We found that this computational difficulty could be overcome if we assume some additional assumptions and aim to achieve a better *generalization error* instead of exactly minimizing the *training error*. The optimization procedure we discuss to obtain such an estimator is the *alternating minimization procedure*, which minimizes the objective function (3) alternately with respect to each component $f_{(r,k)}$. For each component $f_{(r,k)}$, the objective function (3) is a convex function, and thus, it is easy to obtain the optimal solution. Actually, the subproblem is reduced to a variant of the kernel ridge regression, and the solution can be analytically obtained.

The algorithm we call alternating minimization procedure (AMP) is summarized in Algorithm 1. After minimizing the objective (Eq. (4)), the obtained solution is normalized so that its empirical $L_2$-norm becomes 1 to adjust the scaling factor freedom. The parameter $C_n$ in Eq. (4) is a regularization parameter that is appropriately chosen.

For theoretical simplicity, we consider the following equivalent constraint formula instead of the penalization one (4):

$$\tilde{f}'_{(r,k)} \in \operatorname*{argmin}_{\substack{f_{(r,k)} \in \mathcal{H}_{r,k} \\ \|f_{(r,k)}\|_{\mathcal{H}_{r,k}} \leq \tilde{R}}} \left\{ \frac{1}{n} \sum_{i=1}^n \left( y_i - f_{(r,k)}(x_i^{(k)}) \prod_{k' \neq k} \tilde{f}_{(r,k')}(x_i^{(k')}) - \sum_{r' \neq r} \tilde{v}_{r'} \prod_{k'=1}^K \tilde{f}_{(r',k')}(x_i^{(k')}) \right)^2 \right\}$$

$$(5)$$

where the parameter $\tilde{R}$ is a regularization parameter for controlling the complexity of the estimated function.

# 4 Assumptions and problem settings for the convergence analysis

Here, we prepare some assumptions for our theoretical analysis. First, we assume that the distribution $P(X)$ of the input feature $x \in \mathcal{X}$ is a product measure of $P_k(X)$ on each $\mathcal{X}_k$. That is, $P_{\mathcal{X}}(\mathrm{d}X) = P_1(\mathrm{d}X_1) \times \cdots \times P_K(\mathrm{d}X_K)$ for $X = (X_1, \ldots, X_K) \in \mathcal{X} = \mathcal{X}_1 \times \cdots \times \mathcal{X}_K$. This is typically assumed in the analysis of linear tensor estimation methods [13, 6, 3, 21, 1, 36, 27, 37]. Thus, the $L_2$-norm of a "rank-1" function $f(x) = \prod_{k=1}^K f_k(x^{(k)})$ can be decomposed into

$$\|f\|_{L_2(P_{\mathcal{X}})}^2 = \|f_1\|_{L_2(P_1)}^2 \times \cdots \times \|f_K\|_{L_2(P_K)}^2.$$

Hereafter, with a slight abuse of notations, we denote by $\|f\|_{L_2} = \|f\|_{L_2(P_k)}$ for a function $f : \mathcal{X}_k \to \mathbb{R}$. The inner product in the space $L_2$ is denoted by $\langle f, g \rangle_{L_2} := \int f(X)g(X)\mathrm{d}P_{\mathcal{X}}(X)$. Note that because of the construction of $P_{\mathcal{X}}$, it holds that $\langle f, g \rangle_{L_2} = \prod_{k=1}^K \langle f_k, g_k \rangle_{L_2}$ for functions $f(x) = \prod_{k=1}^K f_k(x^{(k)})$ and $g(x) = \prod_{k=1}^K g_k(x^{(k)})$ where $x = (x^{(1)}, \ldots, x^{(K)}) \in \mathcal{X}$.

Next, we assume that the norm of the true function is bounded away from zero and from above. Let the magnitude of the $r$th component of the true function be $v_r := \|\prod_{k=1}^K f_{(r,k)}^*\|_{L_2}$ and the normalized components be $f_{(r,k)}^{**} := f_{(r,k)}^* / \|f_{(r,k)}^*\|_{L_2}$ ($\forall(r,k)$).

**Assumption 1** (Boundedness Assumption)**.**

(A1-1) *There exist $0 < v_{\min} \leq v_{\max}$ such that $v_{\min} \leq v_r \leq v_{\max}$ ($\forall r = 1, \ldots, d$).*

(A1-2) *The true function $f_{(r,k)}^*$ is included in the RKHS $\mathcal{H}_{r,k}$, i.e., $f_{(r,k)}^* \in \mathcal{H}_{r,k}$ ($\forall(r,k)$), and there exists $R > 0$ such that $\max\{v_r, 1\}\|f_{(r,k)}^{**}\|_{\mathcal{H}_{r,k}} \leq R$ ($\forall(r,k)$).*

(A1-3) *The kernel function $\mathrm{k}_{(r,k)}$ associated with the RKHS $\mathcal{H}_{r,k}$ is bounded as $\sup_{x \in \mathcal{X}_k} \mathrm{k}_{(r,k)}(x,x) \leq 1$ ($\forall(r,k)$).*

(A1-4) *There exists $L > 0$ such that the noise is bounded as $|\epsilon_i| \leq L$ (a.s.).*

Assumption 1 is a standard one for the analysis of the tensor model and the kernel regression model. Note that the boundedness condition of the kernel gives that $\|f\|_\infty = \sup_{x^{(k)}} |f(x^{(k)})| \leq \|f\|_{\mathcal{H}_{r,k}}$ for all $f \in \mathcal{H}_{r,k}$ because the Cauchy-Schwarz inequality gives $|\langle f, \mathrm{k}_{(r,k)}(\cdot, x^{(k)})\rangle_{\mathcal{H}_{r,k}}| \leq \mathrm{k}_{(r,k)}(x^{(k)}, x^{(k)})\|f\|_{\mathcal{H}_{r,k}}$ for all $x^{(k)}$. Thus, combining with (A1-2), we also have $\|f_{(r,k)}^{**}\|_\infty \leq R$. The last assumption (A1-4) is a bit restrictive. However, this assumption can be replaced with a Gaussian assumption. In that situation, we may use the Gaussian concentration inequality [17] instead of Talagrand's concentration inequality in the proof.

Next, we characterize the *complexity* of each RKHS $\mathcal{H}_{r,k}$ by using the *entropy number* [33, 25]. The $\epsilon$-*covering number* $\mathcal{N}(\epsilon, \mathcal{G}, L_2(P_{\mathcal{X}}))$ with respect to $L_2(P_{\mathcal{X}})$ is the minimal number of balls with radius $\epsilon$ measured by $L_2(P_{\mathcal{X}})$ needed to cover a set $\mathcal{G} \subset L_2(P_{\mathcal{X}})$. The $i$th entropy number $e_i(\mathcal{G}, L_2(P_{\mathcal{X}}))$ is defined as the infimum of $\epsilon > 0$ such that $\mathcal{N}(\epsilon, \mathcal{G}, L_2) \leq 2^{i-1}$ [25]. Intuitively, if the entropy number is small, the space $\mathcal{G}$ is "simple"; otherwise, it is "complicated."

**Assumption 2** (Complexity Assumption)**.** *Let $\mathcal{B}_{\mathcal{H}_{r,k}}$ be the unit ball of an RKHS $\mathcal{H}_{r,k}$. There exist $0 < s < 1$ and $c$ such that*

$$e_i(\mathcal{B}_{\mathcal{H}_{r,k}}, L_2(P_{\mathcal{X}})) \leq ci^{-\frac{1}{2s}}, \tag{6}$$

*for all $1 \leq r \leq d$ and $1 \leq k \leq K$.*

The optimal rate of the ordinary kernel ridge regression on the RKHS with Assumption 2 is given as $n^{-\frac{1}{1+s}}$ [26]. Next, we give a technical assumption about the $L_\infty$-norm.

**Assumption 3** (Infinity Norm Assumption)**.** *There exist $0 < s_2 \leq 1$ and $c_2$ such that*

$$\|f\|_\infty \leq c_2 \|f\|_{L_2}^{1-s_2} \|f\|_{\mathcal{H}_{r,k}}^{s_2} \quad (\forall f \in \mathcal{H}_{r,k}) \tag{7}$$

*for all $1 \leq r \leq d$ and $1 \leq k \leq K$.*

By Assumption 1, this assumption is always satisfied for $c_2 = 1$ and $s_2 = 1$. $s_2 < 1$ is a nontrivial situation and gives a tighter bound. We would like to note that this condition with $s_2 < 1$ is satisfied

by many practically used kernels such as the Gaussian kernel. In particular, it is satisfied if the kernel is smooth so that $\mathcal{H}_{r,k}$ is included in a Sobolev space $W^{2,s_2}[0,1]$. More formal characterization of this condition using the notion of a *real interpolation space* can be found in [26] and Proposition 2.10 of [5].

Finally, we assume an *incoherence* condition on $\{f^*_{(r,k)}\}_{r,k}$. Roughly speaking, the incoherence property of a set of functions $\{f_{(r,k)}\}_{r,k}$ means that components $\{f_{(r,k)}\}_r$ are linearly independent across different $1 \leq r \leq d$ on the same mode $k$. This is required to distinguish each component. An analogous assumption has been assumed also in the literature of linear models [13, 6, 3, 21, 36, 27].

**Definition 1** (Incoherence)**.** *A set of functions $\{f_{(r,k)}\}_{r,k}$, where $f_{(r,k)} \in L_2(P_k)$, is $\mu$-incoherent if, for all $k = 1, \ldots, K$, it holds that*

$$|\langle f_{(r,k)}, f_{(r',k)} \rangle_{L_2}| \leq \mu \|f_{(r,k)}\|_{L_2} \|f_{(r',k)}\|_{L_2} \ (\forall r \neq r').$$

**Assumption 4** (Incoherence Assumption)**.** *There exists $1 > \mu^* \geq 0$ such that the true function $\{f^*_{(r,k)}\}_{r,k}$ is $\mu^*$-incoherent.*

# 5 Linear convergence of alternating minimization procedure

In this section, we give the convergence analysis of the AMP algorithm. Under the assumptions presented in the previous section, it will be shown that the AMP algorithm shows linear convergence in the sense of optimization algorithm and achieves the minimax optimal rate in the sense of statistical performance. Roughly speaking, if the initial solution is sufficiently close to the true function (namely, in a distance of $O(1)$), then the solution generated by AMP linearly converges to the optimal solution and the estimation accuracy of the final solution is given as $O(dKn^{-\frac{1}{1+s}})$ up to $\log(dK)$ factor.

We analyze how close the updated estimator is to the true one when the $(r,k)$th component is updated from $\tilde{f}_{(r,k)}$ to $\tilde{f}'_{(r,k)}$. The tensor decomposition $\{f_{(r,k)}\}_{r,k}$ of a nonlinear tensor model has a freedom of scaling. Thus, we need to measure the accuracy based on a normalized representation to avoid the scaling factor uncertainty. Let the normalized components of the estimator be $\bar{f}_{(r',k')} = \tilde{f}_{(r',k')}/\|\tilde{f}_{(r',k')}\|_{L_2} \ (\forall (r',k') \in [d] \times [K])$ and $\bar{v}_{r'} = \tilde{v}_{r'} \prod_{k'=1}^{K} \|\tilde{f}_{(r',k')}\|_{L_2} \ (\forall r' \in [d])$. On the other hand, the newly updated $(r,k)$th element is denoted by $\tilde{f}'_{(r,k)}$ (see Eq. (4)) and we denote by $\bar{v}'_r$ the updated value of $\bar{v}_r$ correspondingly: $\bar{v}'_r = \|\tilde{f}'_{(r,k)}\|_{L_2} \prod_{k' \neq k} \|\tilde{f}_{(r,k')}\|_{L_2}$. The normalized newly updated element is denoted by $\bar{f}'_{(r,k)} = \tilde{f}'_{(r,k)}/\|\tilde{f}'_{(r,k)}\|_{L_2}$.

For an estimator $(\bar{f}, \bar{v}) = (\{\bar{f}_{(r',k')}\}_{r',k'}, \{\bar{v}_{r'}\}_{r'})$ which is a couple of the normalized component and the scaling factor, define

$$\mathrm{d}_\infty(\bar{f}, \bar{v}) := \max_{(r',k')} \{v_{r'} \|\bar{f}_{(r',k')} - f^{**}_{(r',k')}\|_{L_2} + |v_{r'} - \bar{v}_{r'}|\}.$$

For any $\lambda_{1,n} > 0$ and $\lambda_{2,n} > 0$ and $\tau > 0$, we let $a_\tau := \max\{1, L\} \max\{1, \tau\} \log(dK)$ and define $\xi_n = \xi_n(\lambda_{1,n}, \tau)$ and $\xi'_n = \xi'_n(\lambda_{2,n}, \tau)$ as [2]

$$\xi_n := a_\tau \left( \frac{K^{\frac{1+2s}{2}} \lambda_{1,n}^{-\frac{s}{2}}}{\sqrt{n}} \vee \frac{K^{\frac{1+2s}{1+s}}}{\lambda_{1,n}^{\frac{2s+(1-s)s_2}{2(1+s)}} n^{\frac{1}{1+s}}} \right), \quad \xi'_n := a_\tau \left( \frac{\lambda_{2,n}^{-\frac{s}{2}}}{\sqrt{n}} \vee \frac{1}{\lambda_{2,n}^{\frac{1}{2}} n^{\frac{1}{1+s}}} \right).$$

**Theorem 2.** *Suppose that Assumptions 1–4 are satisfied, and the regularization parameter $\tilde{R}$ in Eq. (5) is set as $\tilde{R} = 2R$. Let $\hat{R} = 8\tilde{R}/\min\{v_{\min}, 1\}$ and suppose that we have already obtained an estimator $\tilde{f}$ satisfying the following conditions:*

- *The RKHS-norms of $\{\bar{f}_{(r',k')}\}_{r',k'}$ are bounded as $\|\bar{f}_{(r',k')}\|_{\mathcal{H}_{r',k'}} \leq \hat{R}/2 \ (\forall (r',k') \neq (r,k))$.*

- *The distance from the true one is bounded as $\mathrm{d}_\infty(\bar{f}, \bar{v}) \leq \gamma$.*

*Then, for a sufficiently small $\mu^*$ and $\gamma$ (independent of $n$), there exists an event with probability greater than $1 - 3\exp(-\tau)$ where any $(\bar{f}, \bar{v})$ satisfying the above conditions gives*

$$\left( v_r \|\bar{f}'_{(r,k)} - f^{**}_{(r,k)}\|_{L_2} + |\bar{v}'_r - v_r| \right)^2 \leq \frac{1}{2} \mathrm{d}_\infty(\bar{f}, \bar{v})^2 + S_n \hat{R}^{2K} \tag{8}$$

*for any sufficiently large $n$, where $S_n$ is defined for a constant $C'$ depending on $s, s_2, c, c_2$ as*

$$S_n := C' \left[ \xi'_n \lambda_{2,n}^{1/2} + \xi'^2_n + d\xi_n \lambda_{1,n}^{1/2} + \hat{R}^{2(K-1)(\frac{1}{s_2}-1)}(d\xi_n)^{2/s_2}(1+v_{\max})^2 \right].$$

*Moreover, if we denote by $\eta_n$ the right hand side of Eq. (8), then it holds that*

$$\|\bar{f}'_{(r,k)}\|_{\mathcal{H}_{r,k}} \leq \frac{2}{v_r - \sqrt{\eta_n}}\tilde{R}.$$

The proof and its detailed statement are given in the supplementary material (Theorem A.1). It is proven by using such techniques as the so-called peeling device [32] or, equivalently, the local Rademacher complexity [4], and by combining these techniques with the coordinate descent optimization argument. Theorem 2 states that, if the initial solution is sufficiently close to the true one, then the following updated estimator gets closer to the true one and its RKHS-norm is still bounded above by a constant. Importantly, it can be shown that the updated one still satisfies the conditions of Theorem 2 for large $n$. Since the bound given in Theorem 2 is uniform, the inequality (8) can be recursively applied to the sequence of $\hat{f}^{(t)}$ ($t = 1, 2, \dots$).

By substituting $\lambda_{1,n} = K^{-\frac{1+s}{1-s}}d^{-\frac{2}{1-s}}n^{-\frac{1}{1+s}}$ and $\lambda_{2,n} = n^{-\frac{1}{1+s}}$, we have that

$$S_n = O\left(n^{-\frac{1}{1+s}} \vee \left(n^{-\frac{1}{1+s}-(1-s_2)\min\{\frac{1-s}{4(1+s)},\frac{1}{s_2(1+s)}\}}\mathrm{poly}(d,K)\right)\right)\log(dK),$$

where $\mathrm{poly}(d, K)$ means a polynomial of $d, K$. Thus, if $s_2 < 1$ and $n$ is sufficiently large compared with $d$ and $K$, then the second term is smaller than the first term and we have $S_n \leq Cn^{-\frac{1}{1+s}}$ with a constant $C$. Furthermore, we can bound the $L_2$-norm from the true one as in the following theorem.

**Theorem 3.** *Let $(\hat{f}^{(t)}, \hat{v}^{(t)})$ be the estimator at the $t$th iteration. In addition to the assumptions of Theorem 2, suppose that $(\hat{f}^{(1)}, \hat{v}^{(1)})$ satisfies $\mathrm{d}_\infty(\hat{f}^{(1)}, \hat{v}^{(1)})^2 \leq \frac{v_{\min}^2}{8}$ and $S_n\hat{R}^{2K} \leq \frac{v_{\min}^2}{8}$, $s_2 < 1$ and $n \gg d, K$, then $\check{f}^{(t)}(x) = \sum_{r=1}^d \hat{v}_r^{(t)} \prod_{k=1}^K \hat{f}_{(r,k)}^{(t)}(x^{(k)})$ satisfies*

$$\|\check{f}^{(t)} - f^*\|_{L_2}^2 = O\left(dKn^{-\frac{1}{1+s}}\log(dK) + dK\left(3/4\right)^t\right).$$

*for all $t \geq 2$ uniformly with probability $1 - 3\exp(-\tau)$.*

More detailed argument is given in Theorem A.3 in the supplementary material. This means that after $T = O(\log(n))$ iterations, we obtain the estimation accuracy of $O(dKn^{-\frac{1}{1+s}}\log(dK))$. The estimation accuracy bound $O(dKn^{-\frac{1}{1+s}}\log(dK))$ is intuitively natural because we are estimating $d \times K$ functions $\{f_{(r,k)}^*\}_{r,k}$ and the optimal sample complexity to estimate one function $f_{(r,k)}^*$ is known as $n^{-\frac{1}{1+s}}$ [26]. Indeed, recently, it has been shown that this accuracy bound is *minimax optimal* up to $\log(dK)$ factor [14], that is,

$$\inf_{\hat{f}} \sup_{f^*} \mathrm{E}[\|\hat{f} - f^*\|^2] \gtrsim dKn^{-\frac{1}{1+s}}$$

where $\inf$ is taken over all estimators and $\sup$ runs over all low rank tensors $f^*$ with $\|f_{(r,k)}^*\|_{\mathcal{H}_{r,k}} \leq R$. The Bayes estimator also achieves this minimax lower bound [14]. Hence, a rough Bayes estimator would be a good initial solution satisfying the assumptions.

## 6 Relation to existing works

In this section, we describe the relation of our work to existing works. First, our work can be seen as a nonparametric extension of the linear parametric tensor model. The AMP algorithm and related methods for the linear model has been extensively studied in the recent years, e.g. [1, 13, 6, 3, 21, 36, 27, 37]. Overall, the tensor completion problem has been mainly studied instead of a general regression problem. Among the existing works, [37] analyzed the AMP algorithm for a low-rank matrix estimation problem. It is shown that, under an incoherence condition, the AMP algorithm converges to the optimal in a linear rate. However, their analysis is limited to a matrix case. [1] analyzed an alternating minimization approach to estimate a low-rank tensor with positive entries in a noisy observation setting. [13, 6] considered an AMP algorithm for a tensor completion.

Their estimation method is close to our AMP algorithm. However, the analysis is for a linear tensor completion with/without noise and is a different direction from our general nonparametric regression setting. [3, 36] proposed estimation methods other than an alternating minimization one, which were specialized to a linear tensor completion problem.

As for the theoretical analysis for the nonparametric tensor regression model, some Bayes estimators have been analyzed very recently by [14, 12]. They analyzed Bayes methods with Gaussian process priors and showed that the Gaussian process methods possess a good statistical performance. In particular, [14] showed that the Gaussian process method for the nonlinear tensor estimation yields the *mini-max optimality* as an extension of the linear model analysis [28]. However, the Bayes estimators require posterior sampling such as Gibbs sampling, which is rather computationally expensive. On the other hand, the AMP algorithm yields a linear convergence rate and satisfies the minimax optimality. An interesting observation is that the AMP algorithm requires a stronger assumption than the Bayesian one. There would be a trade-off between computational efficiency and statistical property.

## 7  Numerical experiments

We numerically compare the following methods in multitask learning problems (Eq. (2)):

- Gaussian process method (GP-MTL) [14]: The nonparametric Bayesian method with Gaussian process priors. It was shown that the generalization error of GP-MTL achieves the minimax optimal rate [14].
- Our AMP method with different kernels for the latent factors $h_r$ (see Eq. (2)); the Gaussian RBF kernel and the linear kernel. We also examined their mixture like 2 RBF kernels and 1 linear kernel among $d = 3$ components. They are indicated as Lin(1)+RBF(2).

The tensor rank for AMP and GP-MTL was fixed $d = 3$ in the following two data sets. The kernel width and the regularization parameter were tuned by cross validation. We also examined the scaled latent convex regularization method [34]. However, it did not perform well and was omitted.

### 7.1  Restaurant data

Here, we compared the methods in the Restaurant & Consumer Dataset [7]. The task was to predict consumer ratings about several aspects of different restaurants, which is a typical task of a recommendation system. The number of consumers was $M_1 = 138$, and each consumer gave scores of about $M_2 = 3$ different aspects (food quality, service quality, and overall quality). Each restaurant was described by $M_3 = 44$ features as in [20], and the task was to predict the score of an aspect by a certain consumer based on the restaurant feature vector. This is a multitask learning problem consisting of $M_1 \times M_2 = 414$ (nonlinear) regression tasks where the input feature vector is $M_3 = 44$ dimensional. The kernel function representing the task similarities among Task 1 (restaurant) and Task 2 (aspect) are set as $k(p, p') = \delta_{p,p'} + 0.8 \cdot (1 - \delta_{p,p'})$ (where the pair $p, p'$ are restaurants or aspects) [3].

Fig. 1 shows the relative MSE (the discrepancy of MSE from the best one) for different training sample sizes $n$ computed on the validation data against the number of iterations $t$ averaged over 10 repetitions. It can be seen that the validation error dropped rapidly to the optimal one. The best achievable validation error depended on the sample size. An interesting observation was that, until the algorithm converged to the best possible error, it dropped at a linear rate. After it reached the bottom, the error was no longer improved.

Fig. 2 shows the performance comparison between the AMP method with different kernels and the Gaussian process method (GP-MTL). The performances of AMP and GP-MTL were almost identical. Although AMP is computationally quite efficient, as shown in Fig. 1, it did not deteriorate the statistical performance. This is a remarkable property of the AMP algorithm.

### 7.2  Online shopping data

Here, we compared our AMP method with the existing method using data of Yahoo! Japan shopping. Yahoo! Japan shopping contains various types of shops. The dataset is built on the purchase history

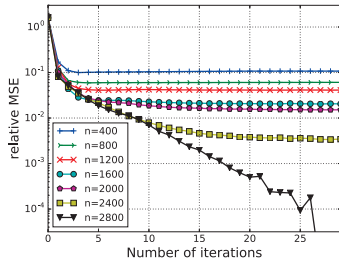
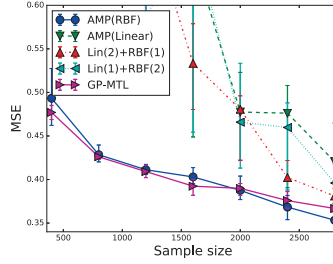
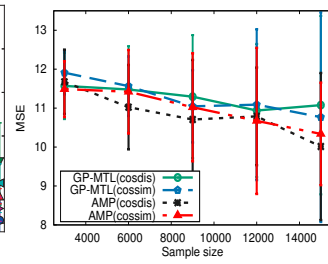

Figure 1: Convergence property of the AMP method: relative MSE against the number of iterations.

Figure 2: Comparison between AMP method with different kernels and GP-MTL on the restaurant data.

Figure 3: Comparison between AMP and GP-MTL on the online shopping data with different kernels.

that describes how many times each consumer bought each product in each shop. Our objective was to predict the quantity of a product purchased by a consumer at a specific shop. Each consumer was described by $65$ features based on his/her properties such as age, gender, and industry type of his/her occupation. We executed the experiments on $100$ items and $508$ different shops. Hence, the problem was reduced to a multitask learning problem consisting of $100 \times 508$ regression tasks.

Similarly to [14], we put a *commute-time* kernel $K = L^{\dagger}$ [8] on the shops based on a Laplacian matrix $L$ on a weighted graph constructed by two similarity measures between shops (where $\dagger$ denotes psuedoinverse). Here, the Lapalacian on the graph is given by $L_{i,j} = \left(\sum_{j \in V} w_{i,j}\right)\delta_{i,j} - w_{i,j}$ where $w_{i,j}$ is the similarity between shops $(i, j)$. We employed the cosine similarity with different parameters as the similarity measures (indicated by "cossim" and "cosdis").

Based on the above settings, we performed a comparison between AMP and GP-MTL with different similarity parameters. We used the Gaussian kernel for the latent factor $h_r$. The result is shown in Fig. 3, which presents the validation error (MSE) against the size of the training data. We can see that, for both "cossim" and "cosdis," AMP performed comparably well to the GP-MTL method and even better than the GP-MTL method in some situations. Here it should be noted that AMP is much more computationally efficient than GP-MTL despite its high predictive performance. This experimental result justifies our theoretical analysis.

# 8 Conclusion

We have developed a convergence theory of the AMP method for the nonparametric tensor learning. The AMP method has been used by several authors in the literature, but its theoretical analysis has not been addressed in the nonparametric setting. We showed that the AMP algorithm converges in a linear rate as an optimization algorithm and achieves the minimax optimal statistical error if the initial point is in the $O(1)$-neighborhood of the true function. We may use the Bayes estimator as a rough initial solution, but it would be an important future work to explore more sophisticated determination of the initial solution.

**Acknowledgment**   This work was partially supported by MEXT kakenhi (25730013, 25120012, 26280009, 15H01678 and 15H05707), JST-PRESTO and JST-CREST.

## Footnotes

[2] The symbol $\vee$ indicates the max operation, that is, $a \vee b := \max\{a, b\}$.

[3]We also tested the delta kernel $k(p, p') = \delta_{p,p'}$, but its performance was worse that the presented kernel.

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
