[Supplementary Material]

# Supplementary material:
# Minimax Optimal Alternating Minimization
# for Kernel Nonparametric Tensor Learning

**Taiji Suzuki**$^*$**, Heishiro Kanagawa**$^\dagger$
$^{*,\dagger}$Department of Mathematical and Computing Science, Tokyo Institute of Technology
$^*$PRESTO, Japan Science and Technology Agency
$^*$Center for Advanced Integrated Intelligence Research, RIKEN
s-taiji@is.titech.ac.jp, kanagawa.h.ab@m.titech.ac.jp

**Hayato Kobayash, Nobuyuki Shimizu, Yukihiro Tagami**
Yahoo Japan Corporation,
hakobaya@yahoo-corp.jp, nobushim@yahoo-corp.jp, yutagami@yahoo-corp.jp

In this supplementary material, we give the comprehensive proof of the theorems in the main text and give more detailed and precise statements of the theorems.

## A    Proof of linear convergence of the alternating minimization procedure

Suppose that we have got an estimator $\tilde{f} = (\tilde{f}_{(r,k)})_{r,k}$, $\tilde{v} = (\tilde{v}_r)_r$ and now we are updating the $(r,k)$-th element as $\tilde{f}'_{(r,k)} \leftarrow \tilde{f}_{(r,k)}$ and $\tilde{v}'_r \leftarrow \tilde{v}_r$.

### A.1    Convergence analysis

Let $\{\widehat{f}_{(r',k')}\}_{r',k'}$ be any functions such that $\prod_{k'=1}^{K} \widehat{f}_{(r',k')} = \tilde{v}_{r'} \prod_{k'=1}^{K} \tilde{f}_{(r',k')}$, and, as a particular choice of such functions, we set $\widehat{f}_{(r',k')} = \tilde{f}_{(r',k')}$ $(\forall k' \neq k, \forall r' \in [d])$ and $\widehat{f}_{(r',k)} = \tilde{v}_{r'} \tilde{f}_{(r',k)}$ $(\forall r' \neq r)$. Let $\bar{f}_{(r',k')} = \tilde{f}_{(r',k')}/\|\tilde{f}_{(r',k')}\|_{L_2} = \widehat{f}_{(r',k')}/\|\widehat{f}_{(r',k')}\|_{L_2}$ $(\forall (r',k') \in [d] \times [K])$ and $\bar{v}_{r'} = \prod_{k'=1}^{K} \|\widehat{f}_{(r',k')}\|_{L_2} = \tilde{v}_{r'} \prod_{k'=1}^{K} \|\tilde{f}_{(r',k')}\|_{L_2}$ $(\forall r' \in [d])$. The newly updated $(r,k)$-th element is denoted by $\tilde{f}'_{(r,k)}$ (see Eq. (4)) and we denote by $\bar{v}'_r$ the updated value of $\bar{v}_r$ correspondingly: $\bar{v}'_r = \|\tilde{f}'_{(r,k)}\|_{L_2} \prod_{k' \neq k} \|\tilde{f}_{(r,k')}\|_{L_2}$. We also denote by $\bar{f}'_{(r,k)} = \tilde{f}'_{(r,k)}/\|\tilde{f}'_{(r,k)}\|_{L_2}$.

For the simplicity of the notation, we denote by $f_r := \prod_{k=1}^{K} f_{(r,k)}$. Similarly, we use notations like $f_r^*$, $\widehat{f}_r$, $\tilde{f}_r$ to express the $r$-th component.

Define $Pf = \int f(X)\mathrm{d}P_{\mathcal{X}}(X)$ and $P_n f := \frac{1}{n}\sum_{i=1}^{n} f(x_i)$ for a function $f : \mathcal{X} \to \mathbb{R}$. For the estimator $\widehat{f} = \{\widehat{f}_{(r,k)}\}_{r,k}$, define

$$\mathrm{d}_\infty(\widehat{f}) := \max_{(r',k')} \{v_{r'}\|\bar{f}_{(r',k')} - f^{**}_{(r',k')}\|_{L_2} + |v_{r'} - \bar{v}_{r'}|\},$$

where $\bar{f}$ and $\bar{v}$ are Note that $\mathrm{d}_\infty(\widehat{f})$ is uniquely defined by $\widehat{f}$. This is equivalent to $\mathrm{d}_\infty(\tilde{f}, \tilde{v})$ in the main text, but we employ the above notation because of the notational simplicity.

For $\lambda_{1,n} > 0$ and $\lambda_{2,n} > 0$, we define

$$\zeta_n = \zeta_n(\lambda_{1,n}) = \max\{C_s, \tilde{C}_s\} \left( \frac{K^{\frac{1+2s}{2}}\lambda_{1,n}^{-\frac{s}{2}}}{\sqrt{n}} \vee \frac{K^{\frac{1+2s}{1+s}}}{\lambda_{1,n}^{\frac{2s+(1-s)s_2}{2(1+s)}} n^{\frac{1}{1+s}}} \right),$$

$$\zeta_n' = \zeta_n'(\lambda_{2,n}) = C_s' \left( \frac{\lambda_{2,n}^{-\frac{s}{2}}}{\sqrt{n}} \vee \frac{1}{\lambda_{2,n}^{\frac{1}{2}} n^{\frac{1}{1+s}}} \right)$$

where $C_s, \tilde{C}_s, C_s'$ are constants depending on $s, s_2, c, c_2$ that will be given in Lemma A.9, Lemma A.11 and Lemma A.15 respectively.

Let $\mathcal{T}_r := \{f - g \mid f = \prod_{k=1}^K f_k, \; g = \prod_{k=1}^K g_k \text{ where } f_k, g_k \in \mathcal{H}_{r,k}, \|f_k\|_{\mathcal{H}_{r,k}} \leq 1, \|g_k\|_{\mathcal{H}_{r,k}} \leq 1 \, (k \in [K])\}$, and $\mathcal{T}_{r,k}' = \{(f_{(r,k)}(x) - f'_{(r,k)}(x)) \prod_{k' \neq k} f_{(r,k')}^{**}(x) \mid f_{(r,k)}, f'_{(r,k)} \in \mathcal{H}_{r,k}, \|f_{(r,k)}\|_{\mathcal{H}_{r,k}} \leq 1, \|f'_{(r,k)}\|_{\mathcal{H}_{r,k}} \leq 1\}$. Then Corollary A.13 and Lemma A.16 yield that there exist universal constants $C$ and $\tilde{C}$ such that

$$\max_{r,r'} \sup_{f \in \mathcal{T}_r, f' \in \mathcal{T}_{r'}} \left| (P - P_n) \left( \frac{f f'}{\|f f'\|_{L_2} + \lambda_{1,n}^{\frac{1}{2}}} \right) \right| \leq C \log(d) \zeta_n \max\{1, \tau\} \tag{S-1}$$

holds with probability $1 - \exp(-\tau)$ for all $\tau > 0$, and all of the following inequalities are simultaneously satisfied with probability $1 - \exp(-\tau)$ for all $\tau > 0$:

$$\max_{1 \leq r \leq d, 1 \leq k \leq K} \sup_{f \in \mathcal{T}_{r,k}'} \left| (P - P_n) \left( \frac{f^2}{\|f^2\|_{L_2} + \lambda_{2,n}^{\frac{1}{2}}} \right) \right| \leq \tilde{C} \log(dK) \zeta_n' \max\{1, \tau\}, \tag{S-2a}$$

$$\max_{1 \leq r \leq d, 1 \leq k \leq K} \sup_{f \in \mathcal{T}_{r,k}'} \left| \frac{1}{n} \sum_{i=1}^n \left( \frac{\epsilon_i f(x_i)}{\|f\|_{L_2} + \lambda_{2,n}^{\frac{1}{2}}} \right) \right| \leq \tilde{C} L \log(dK) \zeta_n' \max\{1, \tau\}, \tag{S-2b}$$

$$\max_{1 \leq r \leq d, 1 \leq k \leq K} \sup_{f \in \mathcal{H}_{r,k}, \|f\|_{\mathcal{H}_{r,k}} \leq 1} \left| (P - P_n) \left( \frac{f^2}{\|f\|_{L_2} + \lambda_{2,n}^{\frac{1}{2}}} \right) \right| \leq \tilde{C} \log(dK) \zeta_n' \max\{1, \tau\}. \tag{S-2c}$$

Let $\tilde{\mathcal{T}}_{r,k} = \{(f_{(r,k)} - f'_{(r,k)})(\prod_{k' \neq k} f_{(r,k')} - \prod_{k' \neq k} f'_{(r,k')}) \mid f_{(r,k')}, f'_{(r,k')} \in \mathcal{H}_{r,k'}, \|f_{(r,k')}\|_{\mathcal{H}_{r,k'}} \leq 1, \|f'_{(r,k')}\|_{\mathcal{H}_{r,k'}} \leq 1 \, (k' \in [K])\}$. Then Lemma A.14 indicates that there exists a universal constant $\tilde{C}' > 0$ such that, for any $0 < \lambda$, all of the following two inequalities simultaneously hold with probability $1 - \exp(-\tau)$:

$$\max_{1 \leq r \leq d, 1 \leq k \leq K} \sup_{f \in \tilde{\mathcal{T}}_{r,k}} \left| \frac{1}{n} \sum_{i=1}^n \left( \frac{\epsilon_i f(x_i)}{\|f\|_{L_2} + \lambda_{1,n}^{\frac{1}{2}}} \right) \right| \leq \tilde{C}' L \log(dK) \zeta_n \max\{1, \tau\}, \tag{S-3a}$$

$$\max_{1 \leq r \leq d, 1 \leq k \leq K} \sup_{f, f' \in \tilde{\mathcal{T}}_{r,k}} \left| (P - P_n) \left( \frac{f f'}{\|f f'\|_{L_2} + \lambda_{1,n}^{\frac{1}{2}}} \right) \right| \leq \tilde{C}' \log(dK) \zeta_n \max\{1, \tau\}. \tag{S-3b}$$

We define an event $\mathcal{E}_1$ so that all inequalities in Eq. (S-1), Eq. (S-2) and Eq. (S-3) are satisfied. Then $P(\mathcal{E}_1) \geq 1 - 3 \exp(-\tau)$ by the argument given above. Based on $\zeta_n(\lambda_{1,n})$ and $\zeta_n'(\lambda_{2,n})$, define

$$\xi_n = \xi_n(\lambda_{1,n}, \tau) := \max\{C, \tilde{C}', \tilde{C}'L\} \log(dK) \zeta_n(\lambda_{1,n}) \max\{1, \tau\},$$

and

$$\xi_n' = \xi_n'(\lambda_{2,n}, \tau) := \tilde{C} \max\{1, L\} \log(dK) \zeta_n'(\lambda_{2,n}) \max\{1, \tau\}.$$

Let $\tilde{R} = 2R$ and $\hat{R} = 8\tilde{R}/\min\{v_{\min}, 1\}$. The following theorem is a detailed version of Theorem 2 in the main text.

**Theorem A.1.** *Suppose that Assumptions 1–4 are satisfied. We also assume the the following conditions.*

- $\mathrm{d}_\infty(\widehat{f})$ *and* $\mu^*$ *are sufficiently small so that there exists* $\mu > 0$ *such that*

$$1 > \mu \geq 2\frac{\mathrm{d}_\infty(\widehat{f})}{v_{\min}} + \mu^*. \tag{S-4}$$

*Correspondingly, we define*

$$c_\mu = (d-1)\left(\frac{4}{3}K + \mu\right)\left(\frac{3}{2}\right)^{K-1}\mu^{K-2}. \tag{S-5}$$

- *Let*

$$Q_n = \frac{2K(1+2\hat{R}^K)\xi_n}{v_{\min}} + 3(d-1)(K-1)\xi_n\hat{R}^K + 4K\hat{R}^K\xi_n + c_\mu + \frac{4\hat{R}K^2}{v_{\min}^2}\mathrm{d}_\infty(\widehat{f}) + \sqrt{\frac{1-s_2}{8}}.$$

*and*

$$\begin{aligned}S_n =&\, 4\xi_n'\lambda_{2,n}^{1/2} + (d+2)\xi_n\lambda_{1,n}^{1/2} + 12\xi_n'^2 \\ &+ s_2\frac{48^{1/s_2}}{8}[(d-1)c_2]^{2/s_2}(1+2v_{\max})^2\hat{R}^{2(K-1)(1-s_2)/s_2}\xi_n^{2/s_2}.\end{aligned}$$

- *n is sufficiently large so that*

$$\xi_n'\hat{R}^2(1+\lambda_{2,n}^{1/2}) \leq \frac{2^{\frac{1}{K}}-1}{2^{1+\frac{1}{K}}-1}.$$

- *The RKHS-norms of the functions* $\{\bar{f}_{(r',k')}\}$ *are bounded as*

$$\|\bar{f}_{(r',k')}\|_{\mathcal{H}_{r',k'}} \leq \frac{1}{2}\hat{R} \quad (\forall (r',k') \neq (r,k)). \tag{S-6}$$

*Then, in the event* $\mathcal{E}_1$, *we have that*

$$\left(v_r\|\bar{f}_{(r,k)}' - f_{(r,k)}^{**}\|_{L_2} + |\bar{v}_r' - v_r|\right)^2 \leq 27Q_n^2\mathrm{d}_\infty(\widehat{f})^2 + 18S_n\hat{R}^{2K}.$$

*In particular, for a sufficiently large* $n$ *and small* $c_\mu$ *such that* $Q_n^2 \leq 1/54$, *we have*

$$\left(v_r\|\bar{f}_{(r,k)}' - f_{(r,k)}^{**}\|_{L_2} + |\bar{v}_r' - v_r|\right)^2 \leq \frac{1}{2}\mathrm{d}_\infty(\widehat{f})^2 + 18S_n\hat{R}^{2K}.$$

*Moreover, if we denote by* $\eta_n$ *the right hand side of the above inequality, then it holds that*

$$\|\bar{f}_{(r,k)}'\|_{\mathcal{H}_{r,k}} \leq \frac{2}{v_r - \sqrt{\eta_n}}\tilde{R}.$$

This theorem immediately gives the following corollary.

**Corollary A.2.** *Let* $\hat{f}_{[t]}$ *be the estimator after the t-th iteration. Suppose that* $\mathrm{d}_\infty(\hat{f}_{[1]})$ *satisfies the assumptions of Theorem A.1,* $Q_n^2 \leq 1/54$, $\mathrm{d}_\infty(\hat{f}_{[1]})^2 \leq v_{\min}^2/8$ *and* $18S_n\hat{R}^{2K} \leq v_{\min}^2/8$. *Then it holds that*

$$\mathrm{d}_\infty(\hat{f}_{[t+1]})^2 \leq \max\left\{\left(\frac{3}{4}\right)^t\mathrm{d}_\infty(\hat{f}_{[1]})^2, 54S_n\hat{R}^{2K}\right\}$$

*for all* $t = 2, 3, \dots$ *in the event* $\mathcal{E}_1$.

By substituting $\lambda_{1,n} = K^{-\frac{1+s}{1-s}}d^{-\frac{2}{1-s}}n^{-\frac{1}{1+s}}$ and $\lambda_{2,n} = n^{-\frac{1}{1+s}}$, we have that

$$S_n = O\left(n^{-\frac{1}{1+s}} \vee \left(n^{-\frac{1}{1+s}-(1-s_2)\min\{\frac{1-s}{4(1+s)}, \frac{1}{s_2(1+s)}\}}\mathrm{poly}(d, K)\right)\right)$$

Thus, for $s_2 < 1$, we have $S_n \leq Cn^{-\frac{1}{1+s}}$ for sufficiently large $n$ with a constant $C$. Since Lemma A.5 states that $\{\bar{f}_{(r,k)}\}_{r,k}$ are $\mu$-incoherent under the assumptions of Theorem A.1, thus Lemma A.7 gives that

$$\|\widehat{f} - f^*\|_{L_2}^2 \leq dK\mathrm{d}_\infty(\widehat{f})^2 + \frac{dK^2}{v_{\min}^2}\mathrm{d}_\infty(\widehat{f})^4 + d^2K^2\mu\mathrm{d}_\infty(\widehat{f})^2.$$

By applying this inequality to $\widehat{f} = \hat{f}_{[t]}$, we obtain the following theorem.

**Theorem A.3.** *In addition to the conditions in Corollary A.2, if* $\mathrm{d}_\infty(\hat{f}_{[1]}) \leq 1/\sqrt{K}$ *and* $\mu \leq 1/(dK)$, *then we have*

$$\|\hat{f}_{[t]} - f^*\|^2 = O\left(dKn^{-\frac{1}{1+s}} \log(dK) + dK(3/4)^t\right),$$

*for sufficiently large* $n$ *and all* $t = 2, 3, \ldots$ *with probability* $1 - 3\exp(-\tau)$.

This means that after $T \geq \frac{1}{\nu}\log(n)$ iterations, we obtain the estimation accuracy $O(dKn^{-\frac{1}{1+s}})$. This computational complexity is quite advantageous. The estimation accuracy is $d \times K$ times the optimal rate $n^{-\frac{1}{1+s}}$ to estimate one function $f^*_{(r,k)}$. This is intuitively natural because we are estimating $d \times K$ functions $\{f^*_{(r,k)}\}_{r,k}$. Indeed, it has been shown that this accuracy bound is minimax optimal.

*Proof.* (Theorem A.1) Throughout the proof, we fix $(r, k)$. There is a freedom of the scaling factor to define $f^*_{(r,k')}$ ($k' = 1, \ldots, K$). Thus, we may set the scaling factor of $f^*$ as

$$f^*_{(r,k')} = f^{**}_{(r,k')}\|\tilde{f}_{(r,k')}\|_{L_2} \text{ for } k' \neq k, \quad f^*_{(r,k)} = v_r f^{**}_{(r,k)} / \prod_{k'\neq k} \|\tilde{f}_{(r,k')}\|_{L_2}.$$

Note that $f^*_r = \prod_{k'=1}^K f^*_{(r,k')} = v_r \prod_{k'=1}^K f^{**}_{(r,k')}$.

Since $\|\tilde{f}_{(r,k')}\|_n = 1$, the $L_2$-norm of $\|\tilde{f}_{(r,k')}\|_{L_2}$ is evaluated as

$$\|\tilde{f}_{(r,k')}\|_{L_2} = \|\tilde{f}_{(r,k')}\|_{L_2} / \|\tilde{f}_{(r,k')}\|_n = \|\bar{f}_{(r,k')}\|_{L_2} / \|\bar{f}_{(r,k')}\|_n = 1/\|\bar{f}_{(r,k')}\|_n. \tag{S-7}$$

By the assumption (S-6) that $\|\bar{f}_{(r,k')}\|_{\mathcal{H}_{r,k'}} \leq \hat{R}$ ($k' \neq k$), Eq. (S-2c) gives that

$$\big|\|\bar{f}_{(r,k')}\|_n^2 - \|\bar{f}_{(r,k')}\|_{L_2}^2\big| \leq \xi'_n \hat{R}^2 (1 + \lambda_{2,n}^{1/2}).$$

By the definition of $\bar{f}_{(r,k')}$, we have $\|\bar{f}_{(r,k')}\|_{L_2} = 1$. Therefore, $\big|\|\bar{f}_{(r,k')}\|_n^2 - 1\big| \leq \xi'_n \hat{R}^2 (1 + \lambda_{2,n}^{1/2})$. Then, by the assumption that $\xi'_n \hat{R}^2 (1 + \lambda_{2,n}^{1/2}) \leq \frac{2^{\frac{1}{K}} - 1}{2^{1+\frac{1}{K}} - 1}$, we have

$$2^{-1/K} \leq \frac{1}{\|\bar{f}_{(r,k')}\|_n^2} \leq 2^{1/K}. \tag{S-8}$$

This and Eq. (S-7) give that $2^{-1/K} \leq \|\tilde{f}_{(r,k')}\|_{L_2} \leq 2^{1/K}$ for $k' \neq k$, and concludes that

$$1/2 \leq \prod_{k'\neq k} \|\tilde{f}_{(r,k')}\|_{L_2} \leq 2.$$

Therefore, by the assumption (A1-2), we have that

$$\|f^*_{(r,k)}\|_{\mathcal{H}_{r,k}} = \frac{v_r \|f^{**}_{(r,k)}\|_{\mathcal{H}_{r,k}}}{\prod_{k'\neq k} \|\tilde{f}_{(r,k')}\|_{L_2}} \leq 2R = \tilde{R}. \tag{S-9}$$

Moreover, by the assumption Eq. (S-6), we have that, for all $k' \neq k$,

$$\|\tilde{f}_{(r,k')}\|_{\mathcal{H}_{r,k'}} = \frac{\|\bar{f}_{(r,k')}\|_{\mathcal{H}_{r,k'}}}{\|\bar{f}_{(r,k')}\|_n} \leq \frac{2^{1/K}}{2} \hat{R} \leq \hat{R}. \tag{S-10}$$

We denote by $F(f)$ the objective function of the optimization problem (4) for the update of $\tilde{f}'_{(r,k)}$ on fixed $(r, k)$. Then by the optimality condition, for the Fréchet derivative $\nabla F(\tilde{f}'_{(r,k)})$ in the RKHS $\mathcal{H}_{r,k}$, it holds that $\langle \nabla F(\tilde{f}'_{(r,k)}), \tilde{f}'_{(r,k)} - f^*_{(r,k)}\rangle_{\mathcal{H}_{r,k}} \leq 0$. That is,

$$\frac{1}{n}\sum_{i=1}^n \left(\tilde{f}'_{(r,k)}(x_i) \prod_{k'\neq k} \tilde{f}_{(r,k')}(x_i) + \sum_{r'\neq r} \tilde{v}_{r'}\tilde{f}_{r'}(x_i) - y_i\right) \prod_{k'\neq k} \tilde{f}_{(r,k')}(x_i)(\tilde{f}'_{(r,k)}(x_i) - f^*_{(r,k)}(x_i))$$

$$\leq 0, \tag{S-11}$$

where we used that $f^*_{(r,k)}$ is in the feasible set because $\|f^*_{(r,k)}\|_{\mathcal{H}_{r,k}} \leq \tilde{R}$ (see Eq. (S-9)). By using the relation $y_i = f^*(x_i) + \epsilon_i$ and arranging the terms in Eq. (S-11), we obtain that

$$
P_n \left[ (\tilde{f}'_{(r,k)} - f^*_{(r,k)})^2 \left( \prod_{k' \neq k} \tilde{f}_{(r,k')} \right)^2 \right]
$$

$$
\leq \frac{1}{n} \sum_{i=1}^{n} \epsilon_i \prod_{k' \neq k} \tilde{f}_{(r,k')}(x_i)(\tilde{f}'_{(r,k)}(x_i) - f^*_{(r,k)}(x_i))
$$

$$
- P_n \left[ \left( \sum_{r' \neq r} (\hat{f}_{r'} - f^*_{r'}) + f^*_{(r,k)} \left( \prod_{k' \neq k} \tilde{f}_{(r,k')} - \prod_{k' \neq k} f^*_{(r,k')} \right) \right) \prod_{k' \neq k} \tilde{f}_{(r,k')} \left( \tilde{f}'_{(r,k)} - f^*_{(r,k)} \right) \right].
$$

(S-12)

Now, let $\tilde{g} := \prod_{k' \neq k} \tilde{f}_{(r,k')} \left( \tilde{f}'_{(r,k)} - f^*_{(r,k)} \right)$, and define $E_1$ to $E_6$ as

$$
E_1 := (P - P_n)(\tilde{g}^2), \quad E_2 := \frac{1}{n} \sum_{i=1}^{n} \epsilon_i \tilde{g}(x_i),
$$

$$
E_3 = (P - P_n) \left[ \left( \sum_{r' \neq r} (\hat{f}_{r'} - f^*_{r'}) \right) \tilde{g} \right], \quad E_4 := (P - P_n) \left[ f^*_{(r,k)} \left( \prod_{k' \neq k} \tilde{f}_{(r,k')} - \prod_{k' \neq k} f^*_{(r,k')} \right) \tilde{g} \right],
$$

$$
E_5 = -P \left[ \sum_{r' \neq r} (\tilde{f}_{r'} - f^*_{r'}) \tilde{g} \right], \quad E_6 := -P \left[ f^*_{(r,k)} \left( \prod_{k' \neq k} \tilde{f}_{(r,k)} - \prod_{k' \neq k} f^*_{(r,k)} \right) \tilde{g} \right].
$$

Then, we can easily see that Eq. (S-12) gives

$$
P(\tilde{g}^2) \leq \sum_{j=1}^{6} E_j \leq |E_1| + |E_2| + |E_3| + |E_4| + |E_5| + |E_6|.
$$

From now on, we are going to bound each term $E_j$ ($j = 1, \ldots, 6$).

(1) (Bounding $E_1$ and $E_2$) Since $\|\tilde{f}_{(r,k')}\|_{\mathcal{H}_{r,k'}} \leq \hat{R}$ ($\forall k' \neq k$) (Eq. (S-10)), $\|f^*_{(r,k)}\|_{\mathcal{H}_{r,k}} \leq \hat{R}$ (Eq. (S-9)), $\|\tilde{f}'_{(r,k)}\|_{\mathcal{H}_{r,k}} \leq \hat{R}$ by the construction, Lemma A.4 gives upper bounds of $|E_1|$ and $|E_2|$ as

$$
|E_1| \leq 2\hat{R}^K \xi'_n \left( \|\tilde{g}\|_{L_2} + \lambda_{2,n}^{1/2} \hat{R}^K \right) + \hat{R}^K \xi_n (2K \|\tilde{g}\|_{L_2} d_\infty(\hat{f})/v_{\min} + \lambda_{1,n}^{1/2} \hat{R}^K),
$$

$$
|E_2| \leq 2\xi'_n \left( \|\tilde{g}\|_{L_2} + \lambda_{2,n}^{1/2} \hat{R}^K \right) + \xi_n (2K \|\tilde{g}\|_{L_2} d_\infty(\hat{f})/v_{\min} + \lambda_{1,n}^{1/2} \hat{R}^K).
$$

(3) (Bounding $E_3$) Eq. (S-1) gives an upper bound of $E_3$ as

$$
|E_3| \leq \sum_{r' \neq r} |(P - P_n)[(\hat{f}_{r'} - f^*_{r'})\tilde{g}]| \leq \sum_{r' \neq r} \xi_n (\|(\hat{f}_{r'} - f^*_{r'})\tilde{g}\|_{L_2} + \lambda_{1,n}^{1/2} \hat{R}^{2K}).
$$

Now we evaluate the term $\|(\tilde{f}_{r'} - f^*_{r'})\tilde{g}\|_{L_2}$. By a slight abuse of notation, we change the scaling of $\tilde{f}$ as $\tilde{f}_{(r',k')} = \bar{f}_{(r',k')}$ ($\forall k' \neq k, r' \neq r$) and $\tilde{f}_{(r',k)} = \|\hat{f}_{r'}\|_{L_2} \bar{f}_{(r',k)} = \bar{v}_{r'} \bar{f}_{(r',k)}$ ($\forall r' \neq r$), in particular, $\hat{f}_{r'} = \prod_{k'=1}^{K} \tilde{f}_{(r',k')}$. Similarly, we set $f^*_{(r',k')} = f^{**}_{(r',k')}$ ($\forall k' \neq k, r' \neq r$) and $f^*_{(r',k)} = v_{r'} f^{**}_{(r',k)}$ ($\forall r' \neq r$). Then, by the assumption (S-6), it holds that

$$
\|\tilde{f}_{(r',k)}\|_{\mathcal{H}_{r',k}} = \bar{v}_{r'} \|\bar{f}_{(r',k)}\|_{\mathcal{H}_{r',k}} \leq \bar{v}_{r'} \frac{\hat{R}}{2} \leq (v_{r'} + d_\infty(\hat{f})) \frac{\hat{R}}{2}. \quad \text{(S-13)}
$$

Hence, the term $\|(\hat{f}_{r'} - f^*_{r'})\tilde{g}\|_{L_2}$ is bounded as

$$
\|(\hat{f}_{r'} - f^*_{r'})\tilde{g}\|_{L_2}^2 = P[(\hat{f}_{r'} - f^*_{r'})^2 \tilde{g}^2]
$$

$$=P\left\{\left[(\tilde{f}_{(r',k)}-f^*_{(r',k)})\prod_{k'\neq k}f^*_{(r',k')}+\tilde{f}_{(r',k)}(\prod_{k'\neq k}\tilde{f}_{(r',k')}-\prod_{k'\neq k}f^*_{(r',k')})\right]^2(\tilde{f}'_{(r,k)}-f^*_{(r,k)})^2(\prod_{k'\neq k}\tilde{f}_{(r,k')})^2\right\}$$

$$\leq 2P\left\{(\tilde{f}_{(r',k)}-f^*_{(r',k)})^2(\prod_{k'\neq k}f^*_{(r',k')})^2(\tilde{f}'_{(r,k)}-f^*_{(r,k)})^2(\prod_{k'\neq k}\tilde{f}_{(r,k')})^2\right.$$

$$\left.+\tilde{f}^2_{(r',k)}(\prod_{k'\neq k}\tilde{f}_{(r',k')}-\prod_{k'\neq k}f^*_{(r',k')})^2(\tilde{f}'_{(r,k)}-f^*_{(r,k)})^2(\prod_{k'\neq k}\tilde{f}_{(r,k')})^2\right\}$$

$$\overset{(a)}{\leq}2\|\tilde{f}_{(r',k)}-f^*_{(r',k)}\|^2_\infty P[\tilde{g}^2(\prod_{k'\neq k}f^*_{(r',k')})^2]+4\bar{v}^2_{r'}\hat{R}^{2K}P(\tilde{g}^2)\|\prod_{k'\neq k}\tilde{f}_{(r',k')}-\prod_{k'\neq k}f^*_{(r',k')}\|^2_{L_2}$$

$$\overset{(b)}{\leq}2\|\tilde{f}_{(r',k)}-f^*_{(r',k)}\|^2_\infty P[\tilde{g}^2(\prod_{k'\neq k}f^*_{(r',k')})^2]+4\hat{R}^{2K}P(\tilde{g}^2)\bar{v}^2_{r'}\left(\sum_{k'\neq k}\|\tilde{f}_{(r',k')}-f^*_{(r',k')}\|_{L_2}\right)^2$$

$$\overset{(c)}{\leq}2\|\tilde{f}_{(r',k)}-f^*_{(r',k)}\|^2_\infty P[\tilde{g}^2(\prod_{k'\neq k}f^*_{(r',k')})^2]+4\hat{R}^{2K}P(\tilde{g}^2)[2(K-1)\mathrm{d}_\infty(\widehat{f})]^2,$$

where the inequalities (a), (b) and (c) are shown as follows: (a) first, we notice that $\|\tilde{f}_{(r',k)}\|_\infty\leq\|\tilde{f}_{(r',k)}\|_{\mathcal{H}_{r',k}}=\bar{v}_{r'}\|\bar{f}_{(r',k)}\|_{\mathcal{H}_{r',k}}\leq\bar{v}_{r'}\hat{R}$ by the assumption (S-6), $\|\tilde{f}_{(r,k')}\|_\infty\leq\|\tilde{f}_{(r,k')}\|_{\mathcal{H}_{r,k'}}\leq\hat{R}$ by Eq. (S-10), and then we obtain

$$P\left\{\tilde{f}^2_{(r',k)}(\prod_{k'\neq k}\tilde{f}_{(r',k')}-\prod_{k'\neq k}f^*_{(r',k')})^2(\tilde{f}'_{(r,k)}-f^*_{(r,k)})^2(\prod_{k'\neq k}\tilde{f}_{(r,k')})^2\right\}$$

$$\leq\|\tilde{f}_{(r',k)}\|^2_\infty\prod_{k'\neq k}\|\tilde{f}_{(r,k')}\|^2_\infty P\left[(\tilde{f}'_{(r,k)}-f^*_{(r,k)})^2\right]P\left[(\prod_{k'\neq k}\tilde{f}_{(r',k')}-\prod_{k'\neq k}f^*_{(r',k')})^2\right]$$

$$\leq\bar{v}^2_{r'}\hat{R}^{2K}P\left[(\tilde{f}'_{(r,k)}-f^*_{(r,k)})^2\prod_{k'\neq k}\tilde{f}^2_{(r,k')}\right]\frac{1}{P(\prod_{k'\neq k}\tilde{f}^2_{(r,k')})}P\left[(\prod_{k'\neq k}\tilde{f}_{(r',k')}-\prod_{k'\neq k}f^*_{(r',k')})^2\right]$$

$$\leq\bar{v}^2_{r'}\hat{R}^{2K}\|\tilde{g}\|^2_{L_2}P\left[(\prod_{k'\neq k}\tilde{f}_{(r',k')}-\prod_{k'\neq k}f^*_{(r',k')})^2\right].$$

(b) is shown by the equalities $\|\tilde{f}_{(r',k')}\|_{L_2}=\|f^*_{(r',k')}\|_{L_2}=1$ and $\bar{v}_{r'}=\|\tilde{f}_{(r',k)}\|_{L_2}$. (c) is shown as $\bar{v}_{r'}\|\bar{f}_{(r',k')}-f^*_{(r',k')}\|_{L_2}=v_{r'}\|\bar{f}_{(r',k')}-f^*_{(r',k')}\|_{L_2}+|\bar{v}_{r'}-v_{r'}|\|\bar{f}_{(r',k')}-f^*_{(r',k')}\|_{L_2}\leq 2(v_{r'}\|\bar{f}_{(r',k')}-f^*_{(r',k')}\|_{L_2}+|\bar{v}_{r'}-v_{r'}|)\leq 2\mathrm{d}_\infty(\widehat{f})$. Here, by Assumption 3 and Eq. (S-13), we have

$$\|\tilde{f}_{(r',k)}-f^*_{(r',k)}\|_\infty$$

$$\leq c_2\|\tilde{f}_{(r',k)}-f^*_{(r',k)}\|^{1-s_2}_{L_2}\|\tilde{f}_{(r',k)}-f^*_{(r',k)}\|^{s_2}_{\mathcal{H}_{r',k}}$$

$$\leq c_2(\|\tilde{f}_{(r',k)}-v_{r'}\bar{f}_{(r',k)}\|_{L_2}+\|v_{r'}\bar{f}_{(r',k)}-f^*_{(r',k)}\|_{L_2})^{1-s_2}\|\tilde{f}_{(r',k)}-f^*_{(r',k)}\|^{s_2}_{\mathcal{H}_{r',k}}$$

$$=c_2(|\bar{v}_{r'}-v_{r'}|\|\bar{f}_{(r',k)}\|_{L_2}+v_{r'}\|\bar{f}_{(r',k)}-f^{**}_{(r,k)}\|_{L_2})^{1-s_2}\|\tilde{f}_{(r',k)}-f^*_{(r',k)}\|^{s_2}_{\mathcal{H}_{r',k}}$$

$$\leq c_2\mathrm{d}_\infty(\widehat{f})^{1-s_2}[\hat{R}+(v_{r'}+\mathrm{d}_\infty(\widehat{f}))\hat{R}]^{s_2}\leq c_2\mathrm{d}_\infty(\widehat{f})^{1-s_2}[(1+2v_{\max})\hat{R}]^{s_2},$$

where the last inequality is shown, by the assumption (S-4), $1\geq\mu\geq2\frac{\mathrm{d}_\infty(\widehat{f})}{v_{\min}}\geq2\frac{\mathrm{d}_\infty(\widehat{f})}{v_{\max}}$.

Therefore, it holds that

$$|E_3|\leq(d-1)\xi_n\left[2c_2(1+2v_{\max})^{s_2}\mathrm{d}_\infty(\widehat{f})^{1-s_2}\hat{R}^{K-1+s_2}\|\tilde{g}\|_{L_2}+4(K-1)\hat{R}^K\mathrm{d}_\infty(\widehat{f})\|\tilde{g}\|_{L_2}+\lambda^{1/2}_{1,n}\hat{R}^{2K}\right].$$

(4) (Bounding $E_4$) Eq. (S-1) gives that

$$|E_4| \leq \xi_n \left( \left\| f^*_{(r,k)} \left( \prod_{k' \neq k} \tilde{f}_{(r,k')} - \prod_{k' \neq k} f^*_{(r,k')} \right) \tilde{g} \right\|_{L_2} + \lambda_{1,n}^{1/2} \hat{R}^{2K} \right).$$

The RHS is bounded as

$$\left\| f^*_{(r,k)} \left( \prod_{k' \neq k} \tilde{f}_{(r,k')} - \prod_{k' \neq k} f^*_{(r,k')} \right) \tilde{g} \right\|_{L_2}$$

$$= \left\| f^*_{(r,k)} (\tilde{f}'_{(r,k)} - f^*_{(r,k)}) \right\|_{L_2} \left\| \left( \prod_{k' \neq k} \tilde{f}_{(r,k')} - \prod_{k' \neq k} f^*_{(r,k')} \right) \prod_{k' \neq k} \tilde{f}_{(r,k')} \right\|_{L_2}$$

$$\leq 2\hat{R} \|\tilde{g}\|_{L_2} \times 2K\hat{R}^{K-1} \mathrm{d}_\infty(\hat{f})/v_{\min},$$

where we used the following relation in the last inequality:

$$\|f^*_{(r,k)}(\tilde{f}'_{(r,k)} - f^*_{(r,k)})\|_{L_2} \leq \|f^*_{(r,k)}\|_\infty \|\tilde{f}'_{(r,k)} - f^*_{(r,k)}\|_{L_2}$$

$$\leq \|f^*_{(r,k)}\|_\infty \|\tilde{f}'_{(r,k)} - f^*_{(r,k)}\|_{L_2} (2\| \prod_{k' \neq k} \tilde{f}_{(r,k')} \|_{L_2})$$

$$= 2\|f^*_{(r,k)}\|_\infty \|(\tilde{f}'_{(r,k)} - f^*_{(r,k)}) \prod_{k' \neq k} \tilde{f}_{(r,k')} \|_{L_2} \leq 2\hat{R}\|\tilde{g}\|_{L_2},$$

and

$$\left\| \left( \prod_{k' \neq k} \tilde{f}_{(r,k')} - \prod_{k' \neq k} f^*_{(r,k')} \right) \prod_{k' \neq k} \tilde{f}_{(r,k')} \right\|_{L_2}$$

$$\leq \hat{R}^{K-1} \left\| \prod_{k' \neq k} \tilde{f}_{(r,k')} - \prod_{k' \neq k} f^*_{(r,k')} \right\|_{L_2}$$

$$= \hat{R}^{K-1} \left\| \prod_{k' \neq k} \tilde{f}_{(r,k')} - \prod_{k' \neq k} f^*_{(r,k')} \right\|_{L_2} \leq \hat{R}^{K-1} \left( \sum_{k' \neq k} \frac{\left\| \tilde{f}_{(r,k')} - f^*_{(r,k')} \right\|_{L_2}}{\|\tilde{f}_{(r,k')}\|_{L_2}} \prod_{k'' \neq k} \|\tilde{f}_{(r,k'')}\|_{L_2} \right)$$

$$\leq 2K\hat{R}^{K-1} \mathrm{d}_\infty(\hat{f})/v_{\min}.$$

Therefore, we have

$$|E_4| \leq 4K\hat{R}^K \xi_n \mathrm{d}_\infty(\hat{f}) \|\tilde{g}\|_{L_2} + \xi_n \lambda_{1,n}^{1/2} \hat{R}^{2K}.$$

(5) Lemma A.5 gives an upper bound of the first term of the RHS as

$$|E_5| = \left| P\left[ \sum_{r' \neq r} (\tilde{f}_{r'} - f^*_{r'})\tilde{g} \right] \right| \leq c_\mu \mathrm{d}_\infty(\hat{f}) \|\tilde{g}\|_{L_2}.$$

(6) Lemma A.6 bounds the second term of the RHS as

$$|E_6| = \left| P\left[ f^*_{(r,k)} \left( \prod_{k' \neq k} \tilde{f}_{(r,k)} - \prod_{k' \neq k} f^*_{(r,k)} \right) \tilde{g} \right] \right| \leq \frac{4\hat{R}K^2}{v_{\min}^2} \|\tilde{g}\|_{L_2} \mathrm{d}_\infty(\hat{f})^2.$$

Combining the results from (1) to (6), we have that

$$P(\tilde{g}^2) \leq 2(1 + \hat{R}^K)\xi_n' \left( \|\tilde{g}\|_{L_2} + \lambda_{2,n}^{1/2} \hat{R}^K \right) + (1 + \hat{R}^K)\xi_n (2K\|\tilde{g}\|_{L_2} \mathrm{d}_\infty(\hat{f})/v_{\min} + \lambda_{1,n}^{1/2} \hat{R}^K)$$

$$+ (d-1)\xi_n \left[ 2c_2(1 + 2v_{\max})^{s_2} \mathrm{d}_\infty(\hat{f})^{1-s_2} \hat{R}^{K-1+s_2} \|\tilde{g}\|_{L_2} + 4(K-1)\mathrm{d}_\infty(\hat{f})\hat{R}^K \|\tilde{g}\|_{L_2} \right]$$

$$
+ \left[ 4K\hat{R}^K \xi_n + c_\mu + \frac{4\hat{R}K^2}{v_{\min}^2} \mathrm{d}_\infty(\hat{f}) \right] \|\tilde{g}\|_{L_2} \mathrm{d}_\infty(\hat{f})
$$

$$
+ [(d-1)\xi_n \lambda_{1,n}^{1/2} + \xi_n \lambda_{1,n}^{1/2}] \hat{R}^{2K}
$$

$$
= 2(1+\hat{R}^K)\xi_n' \|\tilde{g}\|_{L_2} + 2(d-1)\xi_n c_2 (1+2v_{\max})^{s_2} \mathrm{d}_\infty(\hat{f})^{1-s_2} \hat{R}^{K-1+s_2} \|\tilde{g}\|_{L_2}
$$

$$
+ \left[ \frac{2K(1+\hat{R}^K)\xi_n}{v_{\min}} + 4(d-1)(K-1)\xi_n \hat{R}^K + 4K\hat{R}^K \xi_n + c_\mu + \frac{4\hat{R}K^2}{v_{\min}^2} \mathrm{d}_\infty(\hat{f}) \right] \|\tilde{g}\|_{L_2} \mathrm{d}_\infty(\hat{f})
$$

$$
+ [2(1+\hat{R}^K)\xi_n' \lambda_{2,n}^{1/2} + (\hat{R}^K + (d+1)\hat{R}^{2K})\xi_n \lambda_{1,n}^{1/2}].
$$

Then, by using the Cauchy-Schwarz inequality and the Young's inequality,

$$
2(1+\hat{R}^K)\xi_n' \|\tilde{g}\|_{L_2} + 2(d-1)\xi_n c_2 (1+2v_{\max})^{s_2} \mathrm{d}_\infty(\hat{f})^{1-s_2} \hat{R}^{K-1+s_2} \|\tilde{g}\|_{L_2}
$$

$$
\leq \frac{1}{6} \|\tilde{g}\|_{L_2}^2 + 6(1+\hat{R}^K)^2 \xi_n'^2 + \frac{1}{6} \|\tilde{g}\|_{L_2}^2 + 6[(d-1)\xi_n c_2 (1+2v_{\max})^{s_2} \mathrm{d}_\infty(\hat{f})^{1-s_2} \hat{R}^{K-1+s_2}]^2
$$

$$
\leq \frac{1}{3} \|\tilde{g}\|_{L_2}^2 + 6(1+\hat{R}^K)^2 \xi_n'^2
$$

$$
+ \frac{1-s_2}{8} \mathrm{d}_\infty(\hat{f})^2 + s_2 6^{1/s_2} 8^{(1-s_2)/s_2} [(d-1)c_2 (1+2v_{\max})^{s_2} \hat{R}^{K-1+s_2}]^{2/s_2} \xi_n^{2/s_2},
$$

and, we also have

$$
\frac{1-s_2}{8} \mathrm{d}_\infty(\hat{f})^2 +
$$

$$
\left[ \frac{2K(1+2\hat{R}^K)\xi_n}{v_{\min}} + 4(d-1)(K-1)\xi_n \hat{R}^K + 4K\hat{R}^K \xi_n + c_\mu + \frac{4\hat{R}K^2}{v_{\min}^2} \mathrm{d}_\infty(\hat{f}) \right] \|\tilde{g}\|_{L_2} \mathrm{d}_\infty(\hat{f})
$$

$$
\leq \frac{1}{6} \|\tilde{g}\|_{L_2}^2 + \frac{3}{2} Q_n^2 \mathrm{d}_\infty(\hat{f})^2,
$$

where

$$
Q_n = \frac{2K(1+2\hat{R}^K)\xi_n}{v_{\min}} + 4(d-1)(K-1)\xi_n \hat{R}^K + 4K\hat{R}^K \xi_n + c_\mu + \frac{4\hat{R}K^2}{v_{\min}^2} \mathrm{d}_\infty(\hat{f}) + \sqrt{\frac{1-s_2}{8}}.
$$

Moreover, since it holds that

$$
2\frac{(1+\hat{R}^K)}{\hat{R}^{2K}} \xi_n' \lambda_{2,n}^{1/2} + \frac{(\hat{R}^K + (d+1)\hat{R}^{2K})}{\hat{R}^{2K}} \xi_n \lambda_{1,n}^{1/2}
$$

$$
+ 6\frac{(1+\hat{R}^K)^2}{\hat{R}^{2K}} \xi_n'^2 + \frac{1}{\hat{R}^{2K}} s_2 \frac{48^{1/s_2}}{8} [(d-1)c_2 (1+2v_{\max})^{s_2} \hat{R}^{K-1+s_2}]^{2/s_2} \xi_n^{2/s_2}
$$

$$
\leq 4\xi_n' \lambda_{2,n}^{1/2} + (d+2)\xi_n \lambda_{1,n}^{1/2} + 12\xi_n'^2 + s_2 \frac{48^{1/s_2}}{8} [(d-1)c_2 (1+2v_{\max})^{s_2}]^{2/s_2} \hat{R}^{2(K-1)(1-s_2)/s_2} \xi_n^{2/s_2}
$$

$$
=: S_n,
$$

we have that

$$
\|\tilde{g}\|_{L_2}^2 \leq \frac{1}{2} \|\tilde{g}\|_{L_2}^2 + \frac{3}{2} Q_n^2 \mathrm{d}_\infty(\hat{f})^2 + S_n \hat{R}^{2K} \tag{S-14}
$$

$$
\Rightarrow \quad \|\tilde{g}\|_{L_2}^2 \leq 3 Q_n^2 \mathrm{d}_\infty(\hat{f})^2 + 2 S_n \hat{R}^{2K}. \tag{S-15}
$$

The left hand side is lower bounded as follows. Let $\tilde{c} = \prod_{k'\neq k} \|\tilde{f}_{(r,k')}\|_{L_2} (= \prod_{k'\neq k} \|f_{(r,k')}^*\|_{L_2})$. Remind that $\bar{v}_r' = \|\tilde{f}_{(r,k)}'\|_{L_2} \prod_{k'\neq k} \|\tilde{f}_{(r,k')}\|_{L_2}$, $\bar{f}_{(r',k')} = \tilde{f}_{(r',k')}/\|\tilde{f}_{(r',k')}\|_{L_2}$ $(\forall (r',k') \neq (r,k))$ and $\bar{f}_{(r,k)}' = \tilde{f}_{(r,k)}'/\|\tilde{f}_{(r,k)}'\|_{L_2}$. Then $\tilde{c}\tilde{f}_{(r,k)}' = \bar{v}_r' \bar{f}_{(r,k)}'$. Note that $v_r = \prod_{k'=1}^K \|f_{(r,k')}^*\|_{L_2} = \|f_{(r,k)}^*\|_{L_2}\tilde{c}$. Thus,

$$
\|\tilde{g}\|_{L_2}^2 = \|\tilde{f}_{(r,k)}' - f_{(r,k)}^*\|_{L_2}^2 \tilde{c}^2 = \|\bar{v}_r' \bar{f}_{(r,k)}' - v_r f_{(r,k)}^{**}\|_{L_2}^2.
$$

Here, the RHS is lower bounded as

$$\|\bar{v}'_r \bar{f}'_{(r,k)} - v_r f^{**}_{(r,k)}\|^2_{L_2} \geq (\bar{v}'_r)^2 \|\bar{f}'_{(r,k)}\|^2_{L_2} - 2\bar{v}'_r v_r \langle \bar{f}'_{(r,k)}, f^{**}_{(r,k)}\rangle_{L_2} + v_r^2 \|f^{**}_{(r,k)}\|^2_{L_2}$$
$$\geq (\bar{v}'_r)^2 - 2\bar{v}_r v_r + v_r^2 \geq (\bar{v}'_r - v_r)^2.$$

Moreover, we also have another lower bound as

$$\|\bar{v}'_r \bar{f}'_{(r,k)} - v_r f^{**}_{(r,k)}\|_{L_2} = \|(\bar{v}'_r - v_r)\bar{f}'_{(r,k)} + v_r(\bar{f}'_{(r,k)} - f^{**}_{(r,k)})\|_{L_2}$$
$$\geq -\|(\bar{v}'_r - v_r)\bar{f}'_{(r,k)}\|_{L_2} + v_r\|\bar{f}'_{(r,k)} - f^{**}_{(r,k)}\|_{L_2} = -|\bar{v}'_r - v_r| + v_r\|\bar{f}'_{(r,k)} - f^{**}_{(r,k)}\|_{L_2}.$$

Therefore,

$$\|\tilde{g}\|^2_{L_2} \geq \frac{1}{9}\left[v_r\|\bar{f}'_{(r,k)} - f^{**}_{(r,k)}\|_{L_2} + |\bar{v}'_r - v_r|\right]^2. \tag{S-16}$$

Combining Eq. (S-15) and Eq. (S-16), we arrive at

$$\frac{1}{9}\left(v_r\|\bar{f}'_{(r,k)} - f^{**}_{(r,k)}\|_{L_2} + |\bar{v}'_r - v_r|\right)^2 \leq 3Q_n^2 \mathrm{d}_\infty(\widehat{f})^2 + 2S_n \hat{R}^{2K}.$$

This gives the first assertion.

Moreover, since $\bar{f}'_{(r,k)} = \tilde{c}\tilde{f}'_{(r,k)}/\bar{v}'_r$,

$$\|\bar{f}'_{(r,k)}\|_{\mathcal{H}_{r,k}} = \frac{\tilde{c}}{\bar{v}'_r}\|\tilde{f}'_{(r,k)}\|_{\mathcal{H}_{r,k}} \leq \frac{2}{\bar{v}'_r}\|\tilde{f}'_{(r,k)}\|_{\mathcal{H}_{r,k}} \leq \frac{2}{v_r - \sqrt{\eta_n}}\tilde{R}$$

which gives the second assertion.  $\square$

## A.2   Key lemmas

**Lemma A.4.** *Under the same setting as in Theorem A.1, in the event $\mathcal{E}_1$, it holds that*

$$\left|\frac{1}{n}\sum_{i=1}^n \epsilon_i \tilde{g}(x_i)\right| \leq 2\xi'_n\left(\|\tilde{g}\|_{L_2} + \lambda_{2,n}^{1/2}\hat{R}^K\right) + \xi_n(2K\|\tilde{g}\|_{L_2}\mathrm{d}_\infty(\widehat{f})/v_{\min} + \lambda_{1,n}^{1/2}\hat{R}^K).$$

*and*

$$|(P - P_n)(\tilde{g}^2)| \leq 2\hat{R}^K \xi'_n\left(\|\tilde{g}\|_{L_2} + \lambda_{2,n}^{1/2}\hat{R}^K\right) + \hat{R}^K \xi_n(2K\|\tilde{g}\|_{L_2}\mathrm{d}_\infty(\widehat{f})/v_{\min} + \lambda_{1,n}^{1/2}\hat{R}^K).$$

*Proof.* First, note that

$$\frac{1}{n}\sum_{i=1}^n \epsilon_i \tilde{g}(x_i) = \frac{1}{n}\sum_{i=1}^n \epsilon_i \prod_{k'\neq k} f^*_{(r,k')}(x_i)(\tilde{f}'_{(r,k)}(x_i) - f^*_{(r,k)}(x_i))$$

$$+ \frac{1}{n}\sum_{i=1}^n \epsilon_i \left(\prod_{k'\neq k}\tilde{f}_{(r,k')}(x_i) - \prod_{k'\neq k} f^*_{(r,k')}(x_i)\right)(\tilde{f}'_{(r,k)}(x_i) - f^*_{(r,k)}(x_i)). \tag{S-17}$$

Using Eq. (S-2b), the first term is bounded by

$$\xi'_n\left(\|\tilde{f}'_{(r,k)} - f^*_{(r,k)}\|_{L_2} + \lambda_{2,n}^{1/2}\hat{R}^K\right)\prod_{k'\neq k}\|f^*_{(r,k')}\|_{L_2}.$$

Since $\prod_{k'\neq k}\|f^*_{(r,k')}\|_{L_2} = \prod_{k'\neq k}\|\tilde{f}_{(r,k')}\|_{L_2} \leq 2$ by the construction of $f^*_{(r,k')}$, the right hand side is upper bounded by $2\xi'_n(\|\tilde{g}\|_{L_2} + \lambda_{2,n}^{1/2}\hat{R}^K)$.

On the other hand, since Eq. (S-9) and Eq. (S-10) give $\max\{\|f^*_{(r,k')}\|_{\mathcal{H}_{r,k'}}, \|\tilde{f}_{(r,k')}\|_{\mathcal{H}_{r,k'}}\} \leq \hat{R}$ ($\forall k' \neq k$), Eq. (S-3a) gives an upper bound of the second term as

$$\xi_n\left(\|\prod_{k'\neq k}\tilde{f}_{(r,k')} - \prod_{k'\neq k} f^*_{(r,k')}\|_{L_2}\|\tilde{f}'_{(r,k)} - f^*_{(r,k)}\|_{L_2} + \lambda_{1,n}^{1/2}\hat{R}^K\right).$$

Since $\|\tilde{f}_{(r,k')}\|_{L_2} = \|f^*_{(r,k')}\|_{L_2} \geq 1/2 \ (\forall k' \neq k)$, it holds that

$$\|\prod_{k' \neq k} \tilde{f}_{(r,k')} - \prod_{k' \neq k} f^*_{(r,k')}\|_{L_2}$$

$$\leq \sum_{k' \neq k} \left( \prod_{l < k', l \neq k} \|\tilde{f}_{(r,l)}\|_{L_2} \right) \|\tilde{f}_{(r,k')} - f^*_{(r,k')}\|_{L_2} \left( \prod_{l > k', l \neq k} \|f^*_{(r,l)}\|_{L_2} \right)$$

$$\leq 2\|\prod_{k' \neq k} \tilde{f}_{(r,k')}\|_{L_2} K \mathrm{d}_\infty(\widehat{f})/v_{\min}.$$

This and $\|g\|_{L_2} = \|\tilde{f}'_{(r,k)} - f^*_{(r,k)}\|_{L_2} \prod_{k' \neq k} \|\tilde{f}_{(r,k')}\|_{L_2}$ give a bound of Eq. (S-17) as

$$\xi_n (2K\|\tilde{g}\|_{L_2} \mathrm{d}_\infty(\widehat{f})/v_{\min} + \lambda_{1,n}^{1/2} \hat{R}^K).$$

The second assertion is also proven by the similar argument to the first assertion by noticing

$$\left( \prod_{k' \neq k} \tilde{f}^2_{(r,k')} - \prod_{k' \neq k} f^{*2}_{(r,k')} \right) (\tilde{f}'_{(r,k)} - f^*_{(r,k)})^2$$

$$= \left( \prod_{k' \neq k} \tilde{f}_{(r,k')} - \prod_{k' \neq k} f^*_{(r,k')} \right) (\tilde{f}'_{(r,k)} - f^*_{(r,k)}) \times \left( \prod_{k' \neq k} \tilde{f}_{(r,k')} + \prod_{k' \neq k} f^*_{(r,k')} \right) (\tilde{f}'_{(r,k)} - f^*_{(r,k)}),$$

and applying Eq. (S-2a) and Eq. (S-3b) instead of Eq. (S-2b) and Eq. (S-3a). □

**Lemma A.5.** *Suppose that the Incoherent Assumption 4 is satisfied. Then, if $\{\tilde{f}_{(r,k)}\}$ and $\mu^*$ satisfy Eq. (S-5), then we have that*

$$P\left[ \sum_{r' \neq r} (\tilde{f}_{r'} - f^*_{r'})\tilde{g} \right] \leq c_\mu \mathrm{d}_\infty(\widehat{f})\|\tilde{g}\|_{L_2}. \tag{S-18}$$

*Moreover, $\{\tilde{f}_{(r,k)}\}_{r,k}$ are $\mu$-incoherent where $\mu = 2\frac{\mathrm{d}_\infty(\widehat{f})}{v_{\min}} + \mu^*$.*

*Proof.* First we show that $\{\tilde{f}_{(r,k)}\}_{r,k}$ are $\mu$-incoherent. This can be shown that

$$|\langle \bar{f}_{(r',k')}, \bar{f}_{(r'',k'')} \rangle|$$
$$= |\langle \bar{f}_{(r',k')} - f^{**}_{(r',k')} + f^{**}_{(r',k')}, \bar{f}_{(r'',k'')} \rangle|$$
$$= |\langle \bar{f}_{(r',k')} - f^{**}_{(r',k')}, \bar{f}_{(r'',k'')} \rangle| + |\langle f^{**}_{(r',k')}, \bar{f}_{(r'',k'')} - f^{**}_{(r'',k'')} + f^{**}_{(r'',k'')} \rangle|$$
$$\leq \|\bar{f}_{(r',k')} - f^{**}_{(r',k')}\|_{L_2} \|\bar{f}_{(r'',k'')}\|_{L_2} + \|f^{**}_{(r',k')}\|_{L_2} \|\bar{f}_{(r'',k'')} - f^{**}_{(r'',k'')}\| + |\langle f^{**}_{(r',k')}, f^{**}_{(r'',k'')} \rangle|$$
$$\leq 2\frac{\mathrm{d}_\infty(\widehat{f})}{v_{\min}} + \mu^* \leq \mu.$$

Let $\Delta f_{(r',k')} = \bar{f}_{(r',k')} - f^{**}_{(r',k')}$ for $k' \neq k$ and $\Delta f_{(r',k)} = \bar{v}_{r'} \bar{f}_{(r',k)} - v_{r'} f^{**}_{(r',k)}$. Then, for $k' \neq k$, $\bar{v}_{r'} \|\Delta f_{(r',k')}\|_{L_2} = v_{r'} \|\Delta f_{(r',k')}\|_{L_2} + |v_{r'} - \bar{v}_{r'}| \|\Delta f_{(r',k')}\|_{L_2} \leq v_{r'} \|\Delta f_{(r',k')}\|_{L_2} + 2|v_{r'} - \bar{v}_{r'}| \leq 2\mathrm{d}_\infty(\tilde{f})$, and $\|\Delta f_{(r',k)}\|_{L_2} \leq \|v_{r'} \bar{f}_{(r',k)} - v_{r'} f^{**}_{(r',k)}\|_{L_2} + |v_{r'} - \bar{v}_{r'}| \leq \mathrm{d}_\infty(\widehat{f})$. Therefore, for sufficiently small $\mathrm{d}_\infty(\widehat{f})$, the LHS of Eq. (S-18) is bounded by

$$P\left[ \sum_{r' \neq r} (\tilde{f}_{r'} - f^*_{r'})\tilde{g} \right] \leq P\left[ \sum_{r' \neq r} [\tilde{f}_{r'} - (\bar{v}_{r'} \bar{f}_{(r',k)} - \Delta f_{(r',k)}) \prod_{k' \neq k} (\bar{f}_{(r',k')} - \Delta f_{(r',k')})]\tilde{g} \right]$$

$$\leq \left| P\left[ \sum_{r' \neq r} \Delta f_{(r',k)} (\tilde{f}'_{(r,k)} - f^*_{(r,k)}) \left( \prod_{k' \neq k} \tilde{f}_{(r,k')} \right) \left( \prod_{k'' \neq k} (\bar{f}_{(r',k'')} - \Delta f_{(r',k'')}) \right) \right] \right|$$

$$+ \left| P\left[ \sum_{r' \neq r} (\tilde{f}'_{(r,k)} - f^*_{(r,k)})\bar{v}_{r'} \bar{f}_{(r',k)} \left( \prod_{k'' \neq k} \tilde{f}_{(r,k'')} \right) \left( \prod_{k'' \neq k} (\bar{f}_{(r',k'')} - \Delta f_{(r',k'')}) - \prod_{k'' \neq k} \bar{f}_{(r',k'')} \right) \right] \right|$$

$$\leq \sum_{r' \neq r} \|\tilde{f}'_{(r,k)} - f^*_{(r,k)}\|_{L_2} \|\Delta f_{(r',k)}\|_{L_2} \prod_{k' \neq k} \|\tilde{f}_{(r,k')}\|_{L_2} \prod_{k'' \neq k} (\mu \|\bar{f}_{(r',k')}\|_{L_2} + \|\Delta f_{(r',k'')}\|_{L_2})$$

$$+ \sum_{r' \neq r} \sum_{k' \neq k} \|\tilde{f}'_{(r,k)} - f^*_{(r,k)}\|_{L_2} \bar{v}_{r'} \|\bar{f}_{(r',k)}\|_{L_2} \|\Delta f_{(r',k')}\|_{L_2}$$

$$\times \prod_{k'' \neq k} \|\tilde{f}_{(r,k'')}\|_{L_2} \prod_{k''' \neq k, k'} (\mu \|\bar{f}_{(r',k'')}\|_{L_2} + \|\Delta f_{(r',k''')}\|_{L_2})$$

$$\overset{(a)}{\leq} (d-1)\|\tilde{f}'_{(r,k)} - f^*_{(r,k)}\|_{L_2} \prod_{k' \neq k} \|\tilde{f}_{(r,k')}\|_{L_2} \mathrm{d}_\infty(\widehat{f}) \left[ \left( \mu + \frac{\mathrm{d}_\infty(\widehat{f})}{v_{\min}} \right)^{K-1} + 2K \left( \mu + \frac{\mathrm{d}_\infty(\widehat{f})}{v_{\min}} \right)^{K-2} \right]$$

$$\overset{(b)}{\leq} \|\tilde{g}\|_{L_2} \mathrm{d}_\infty(\widehat{f})(d-1) \left[ \left( \mu + \frac{\mu}{2} \right)^{K-1} + 2K \left( \mu + \frac{\mu}{2} \right)^{K-2} \right]$$

$$\overset{(c)}{\leq} c_\mu \|\tilde{g}\|_{L_2} \mathrm{d}_\infty(\widehat{f}),$$

where, in the inequality (a), we used the relation $\|\Delta f_{(r',k)}\|_{L_2} \leq \mathrm{d}_\infty(\widehat{f})$, $\bar{v}_{r'} \|\Delta f_{(r',k')}\|_{L_2} \leq 2\mathrm{d}_\infty(\widehat{f})$ for $k' \neq k$, and $\|\Delta f_{(r',k')}\|_{L_2} \leq \frac{\mathrm{d}_\infty(\widehat{f})}{v_{\min}}$ for $k' \neq k$; in the inequality (b), we used the assumption on $\mu$ and $\mathrm{d}_\infty(\widehat{f})$; and, in the final inequality (c), we used the definition of $c_\mu$.

$\square$

**Lemma A.6.** *If* $\prod_{k' \neq k} \|\tilde{f}_{(r,k')}\|_{L_2} \geq 1/2$, *then*

$$P \left[ f^*_{(r,k)} \Big( \prod_{k' \neq k} \tilde{f}_{(r,k')} - \prod_{k' \neq k} f^*_{(r,k')} \Big) \tilde{g} \right] \leq \frac{2\hat{R}K^2}{v_{\min}^2} \|\tilde{g}\|_{L_2} \mathrm{d}_\infty(\widehat{f})^2.$$

*Proof.* Because $P_{\mathcal{X}}$ is a product measure given by $P_{\mathcal{X}} = P_1 \times \cdots \times P_K$, we have that

$$P \left[ f^*_{(r,k)} \Big( \prod_{k' \neq k} \tilde{f}_{(r,k')} - \prod_{k' \neq k} f^*_{(r,k')} \Big) \tilde{g} \right] = P \left[ f^*_{(r,k)} (\tilde{f}_{(r,k)} - f^*_{(r,k)}) \Big( \prod_{k' \neq k} \tilde{f}_{(r,k')} - \prod_{k' \neq k} f^*_{(r,k')} \Big) \prod_{k' \neq k} \tilde{f}_{(r,k')} \right]$$

$$= P \left[ f^*_{(r,k)} (\tilde{f}_{(r,k)} - f^*_{(r,k)}) \right] \times P \left[ \Big( \prod_{k' \neq k} \tilde{f}_{(r,k')} - \prod_{k' \neq k} f^*_{(r,k')} \Big) \prod_{k' \neq k} \tilde{f}_{(r,k')} \right]$$

$$\leq \|f^*_{(r,k)}\|_\infty \|\tilde{f}_{(r,k)} - f^*_{(r,k)}\|_{L_2} P \left[ \prod_{k' \neq k} \tilde{f}^2_{(r,k')} - \prod_{k' \neq k} f^*_{(r,k')} \prod_{k' \neq k} \tilde{f}_{(r,k')} \right],$$

where we used the Cauchy-Schwarz inequality in the second line. Here, by the construction of $\widehat{f}_{(r,k)}$, we have that $\| \prod_{k' \neq k} \tilde{f}_{(r,k')}\|_{L_2} = \| \prod_{k' \neq k} f^*_{(r,k')}\|_{L_2}$ and thus

$$P \left[ \prod_{k' \neq k} \tilde{f}_{(r,k')} \prod_{k' \neq k} \tilde{f}_{(r,k')} - \prod_{k' \neq k} f^*_{(r,k')} \prod_{k' \neq k} \tilde{f}_{(r,k')} \right] = \frac{1}{2} \left\| \prod_{k' \neq k} \tilde{f}_{(r,k')} - \prod_{k' \neq k} f^*_{(r,k')} \right\|^2_{L_2}.$$

Here, since it holds that

$$\left\| \prod_{k' \neq k} \tilde{f}_{(r,k')} - \prod_{k' \neq k} f^*_{(r,k')} \right\|_{L_2}$$

$$\leq \sum_{k' \neq k} \left( \|\tilde{f}_{(r,k')} - f^*_{(r,k')}\|_{L_2} \prod_{k'' < k', k'' \neq k} \|\tilde{f}_{(r,k'')}\|_{L_2} \prod_{k'' > k', k'' \neq k} \|f^*_{(r,k')}\|_{L_2} \right)$$

$$\leq 2^{\frac{K-1}{K}} \sum_{k' \neq k} \|\tilde{f}_{(r,k')} - f^*_{(r,k')}\|_{L_2} / \|\tilde{f}_{(r,k')}\|_{L_2} \leq 2K \mathrm{d}_\infty(\widehat{f})/v_{\min},$$

we have that

$$P \left[ f^*_{(r,k)} \Big( \prod_{k' \neq k} \tilde{f}_{(r,k')} - \prod_{k' \neq k} f^*_{(r,k')} \Big) \tilde{g} \right] \leq \|f^*_{(r,k)}\|_\infty \|\tilde{f}_{(r,k)} - f^*_{(r,k)}\|_{L_2} 2K^2 \mathrm{d}_\infty(\widehat{f})^2 / v_{\min}^2.$$

Moreover, since $\|f^*_{(r,k)}\|_\infty \le \|f^*_{(r,k)}\|_{\mathcal{H}_{r,k}} \le \hat{R}$ (Eq. (S-9)) and $\prod_{k'\neq k}\|\tilde{f}_{(r,k')}\|_{L_2} \ge 1/2$ gives $\|\tilde{f}_{(r,k)} - f^*_{(r,k)}\|_{L_2} \le 2\|\tilde{g}\|_{L_2}$, we have that

$$P\left[f^*_{(r,k)}\big(\prod_{k'\neq k}\tilde{f}_{(r,k)} - \prod_{k'\neq k}f^*_{(r,k)}\big)\tilde{g}\right] \le \frac{4\hat{R}K^2}{v_{\min}^2}\|\tilde{g}\|_{L_2}d_\infty(\hat{f})^2.$$

$\square$

**Lemma A.7.** *If $\{\hat{f}_{(r,k)}\}_{r,k}$ and $\{f^*_{(r,k)}\}_{r,k}$ are $\mu$-incoherent, then we have*

$$\|\hat{f} - f^*\|_{L_2}^2 \le dK d_\infty(\hat{f})^2 + \frac{dK^2}{v_{\min}^2}d_\infty(\hat{f})^4 + d^2K^2\mu d_\infty(\hat{f})^2.$$

*Proof.*

$$\|\hat{f} - f^*\|_{L_2}^2 = \|\sum_{r=1}^{d}\sum_{k=1}^{K}\prod_{l<k}\hat{f}_{(r,l)}(\hat{f}_{(r,k)} - f^{**}_{(r,k)})\prod_{l>k}f^{**}_{(r,l)}\|_{L_2}^2$$

Let

$$\Delta\hat{f}_{(r,k)} := \prod_{l<k}\hat{f}_{(r,l)}(\hat{f}_{(r,k)} - f^{**}_{(r,k)})\prod_{l>k}f^{**}_{(r,l)}.$$

Now we set $\hat{f}_{(r,k)} = \bar{f}_{(r,k)}$ ($\forall r \in [d], k \in [K-1]$), $\hat{f}_{(r,K)} = \bar{v}_r\bar{f}_{(r,K)}$ ($\forall r \in [d]$), $f^*_{(r,k)} = f^{**}_{(r,k)}$ ($\forall r \in [d], k \in [K-1]$), and $f^*_{(r,K)} = v_r f^{**}_{(r,K)}$ ($\forall r \in [d]$). Then, it holds that, for all $r \in [d]$,

$$\Delta\hat{f}_{(r,k)} = v_r\prod_{l<k}\bar{f}_{(r,l)}(\bar{f}_{(r,k)} - f^{**}_{(r,k)})\prod_{l>k}f^{**}_{(r,l)} \quad (\forall k < K),$$

$$\Delta\hat{f}_{(r,K)} = \prod_{l<K}\bar{f}_{(r,l)}(\bar{v}_r\bar{f}_{(r,K)} - v_r f^{**}_{(r,K)}).$$

By the definition of $\Delta\hat{f}_{(r,k)}$, we have that

$$(\hat{f} - f^*)^2 = (\sum_{r=1}^{d}\sum_{k=1}^{K}\Delta\hat{f}_{(r,k)})^2$$

$$= \sum_{r=1}^{d}\left(\sum_{k=1}^{K}\Delta\hat{f}_{(r,k)}^2 + \sum_{k\neq k'}\Delta\hat{f}_{(r,k)}\Delta\hat{f}_{(r,k')}\right) + \sum_{r\neq r'}\sum_{k=1}^{K}\sum_{k'=1}^{K}\Delta\hat{f}_{(r,k)}\Delta\hat{f}_{(r',k')}.$$

We evaluate each term. If $k < K$, we have

$$P(\Delta\hat{f}_{(r,k)}^2) = v_r^2\|\bar{f}_{(r,k)} - f^{**}_{(r,k)}\|_{L_2}^2 \le d_\infty(\hat{f})^2,$$

otherwise, we have

$$P(\Delta\hat{f}_{(r,K)}^2) = \|\bar{v}_r\bar{f}_{(r,K)} - v_r f^{**}_{(r,K)}\|_{L_2}^2 \le \left(\bar{v}_r\|\bar{f}_{(r,K)} - f^{**}_{(r,K)}\|_{L_2} + |\bar{v}_r - v_r|\|\bar{f}_{(r,K)}\|_{L_2}\right)^2$$

$$\le d_\infty(\hat{f})^2.$$

Next we evaluate the term $\Delta\hat{f}_{(r,k)}\Delta\hat{f}_{(r,k')}$ with $k \neq k'$. If $k < k' < K$, then

$$P(\Delta\hat{f}_{(r,k)}\Delta\hat{f}_{(r,k')})$$
$$\le v_r^2|P[(\bar{f}_{(r,k)} - f^{**}_{(r,k)})\bar{f}_{(r,k)}]P[f^{**}_{(r,k')}(\bar{f}_{(r,k')} - f^{**}_{(r,k')})]|$$
$$= v_r^2|(1 - P[f^{**}_{(r,k)}\bar{f}_{(r,k)}])(P[f^{**}_{(r,k')}(\bar{f}_{(r,k')}] - 1)| \quad (\because \|f^{**}_{(r,k)}\|_{L_2} = \|\bar{f}_{(r,k)}\|_{L_2} = 1)$$
$$= v_r^2\frac{1}{4}|P[(f^{**}_{(r,k)})^2 - 2f^{**}_{(r,k)}\bar{f}_{(r,k)} + (\bar{f}_{(r,k)})^2]||P[2f^{**}_{(r,k')}\bar{f}_{(r,k')} - (f^{**}_{(r,k')})^2 - (\bar{f}_{(r,k')})^2]|$$
$$(\because \|f^{**}_{(r,k)}\|_{L_2} = \|\bar{f}_{(r,k)}\|_{L_2} = 1)$$

$$= v_r^2 \|f_{(r,k)}^{**} - \bar{f}_{(r,k)}\|_{L_2}^2 \|f_{(r,k')}^{**} - \bar{f}_{(r,k')}\|_{L_2}^2$$
$$\leq \mathrm{d}_\infty(\widehat{f})^4 / v_{\min}^2.$$

On the other hand, if $k < k' = K$, then, with a similar argument, we have

$$P(\Delta \widehat{f}_{(r,k)} \Delta \widehat{f}_{(r,k')}) \leq \mathrm{d}_\infty(\widehat{f})^4 / v_{\min}^2.$$

Finally, we evaluate the term $\Delta \widehat{f}_{(r,k)} \Delta \widehat{f}_{(r',k')}$ with $r \neq r'$ ($k$ and $k'$ could be same). If $1 < k, k' < K$, we have

$$P(\Delta \widehat{f}_{(r,k)} \Delta \widehat{f}_{(r',k)})$$
$$\leq v_r v_{r'} |(P \bar{f}_{(r,1)} \bar{f}_{(r',1)})| \times \|\bar{f}_{(r,k)} - f_{(r,k)}^{**}\|_{L_2} \|\bar{f}_{(r',k')} - f_{(r',k')}^{**}\|_{L_2} \leq \mu \mathrm{d}_\infty(\widehat{f})^2,$$

else, we also have the same upper bound.

Combining these inequalities, we have that

$$\|\widehat{f} - f^*\|_{L_2}^2 \leq dK \mathrm{d}_\infty(\widehat{f})^2 + \frac{dK^2}{v_{\min}^2} \mathrm{d}_\infty(\widehat{f})^4 + d^2 K^2 \mu \mathrm{d}_\infty(\widehat{f})^2.$$

$\square$

## A.3  Technical lemmas

Here we give some technical lemmas to show the main theorem (Theorem A.1) .

We denote by $\{\sigma_i\}_{i=1}^n$ the Rademacher random variable that is an i.i.d. random variable such that $\sigma_i \in \{\pm 1\}$. It is known that, for a set of measurable functions $\mathcal{F}$ that is separable with respect to $\infty$-norm, the *Rademacher complexity* $\mathrm{E}[\sup_{f \in \mathcal{F}} \frac{1}{n} \sum_{i=1}^n \sigma_i f(x_i)]$ of $\mathcal{F}$ bounds the supremum of the discrepancy between the empirical and population means of all functions $f \in \mathcal{F}$ (see Lemma 2.3.1 of [8]):

$$\mathrm{E}\left[\sup_{f \in \mathcal{F}} \left| \frac{1}{n} \sum_{i=1}^n (f(x_i) - \mathrm{E}[f]) \right| \right] \leq 2\mathrm{E}\left[\sup_{f \in \mathcal{F}} \left| \frac{1}{n} \sum_{i=1}^n \sigma_i f(x_i) \right| \right], \tag{S-19}$$

where the expectations are taken for both $\{x_i\}_{i=1}^n$ and $\{\sigma_i\}_{i=1}^n$.

The following proposition is the key in our analysis.

**Proposition A.8.** *Let $\mathcal{B}_{\delta,a,b} \subset L_2(P_{\mathcal{X}})$ be a set such that $\forall f \in \mathcal{B}_{\delta,a,b}$ satisfies $\|f\|_{L_2} \leq \delta, \|f\|_\infty \leq b$, and it has a complexity bound like Assumption 2 such that*

$$e_i(\mathcal{B}_{\delta,a,b}, L_2(P_{\mathcal{X}})) \leq a i^{-\frac{1}{2s}}.$$

*Then, there exist constants $C_s'$ depending only on $s$ such that*

$$\mathrm{E}\left[\sup_{f \in \mathcal{B}_{\delta,a,b}} \left| \frac{1}{n} \sum_{i=1}^n \sigma_i f(x_i) \right| \right] \leq C_s' \left( \frac{\delta^{1-s} a^s}{\sqrt{n}} \vee a^{\frac{2s}{1+s}} b^{\frac{1-s}{1+s}} n^{-\frac{1}{1+s}} \right).$$

*Proof.* The proof is given by combining Theorem 7.16 and Corollary 7.31 of [4].  $\square$

Using Proposition A.8 and the *peeling device* [7], we obtain the following lemma (see also [3, 5]).

**Lemma A.9.** *Under the Complexity Assumption (Assumption 2) and the Infinity-Norm Assumption (Assumption 3), there exists a constant $C_s$ depending only on $s, s_2$ and $c, c_2$ such that for all $\lambda > 0$*

$$\mathrm{E}\left[ \sup_{f_{(r,k)} \in \mathcal{H}_{r,k}: \|f_{(r,k)}\|_{\mathcal{H}_{r,k}} \leq 1} \frac{|\frac{1}{n} \sum_{i=1}^n \sigma_i \prod_{k=1}^K f_{(r,k)}(x_i)|}{\prod_{k=1}^K \|f_{(r,k)}\|_{L_2} + \lambda^{\frac{1}{2}}} \right] \leq C_s \left( \frac{K^{\frac{1+2s}{2}} \lambda^{-\frac{s}{2}}}{\sqrt{n}} \vee \frac{K^{\frac{1+2s}{1+s}} \lambda^{-\frac{2s+(1-s)s_2}{2(1+s)}}}{n^{\frac{1}{1+s}}} \right).$$

*Proof.* (Lemma A.9) Let $\mathcal{H}_{r,k}(\delta) := \{f \in \mathcal{H}_{r,k} \mid \|f\|_{\mathcal{H}_{r,k}} \leq 1, \|f\|_{L_2} \leq \delta\}$ and $z = 2^{1/s} > 1$. We evaluate the entropy number of the set $\mathcal{B} = \{\prod_{k=1}^K f_k \mid f_k \in \mathcal{H}_{r,k}(\delta_k)\}$. For all $f \in \mathcal{B}$, we have

$\|f\|_{L_2} \leq \prod_{k=1}^{K} \delta_k$ because for $f = \prod_{k=1}^{K} f_k$ it holds that $\|f\|_{L_2} = \prod_{k=1}^{K} \|f_k\|_{L_2}$. Moreover, since $\|f_k\|_\infty \leq \|f_k\|_{\mathcal{H}_{r,k}}$ ($f_k \in \mathcal{H}_{r,k}$), we have $\|f\|_\infty \leq 1$ for all $f \in \mathcal{B}$. The $L_2(P_\mathcal{X})$ norm between $f = \prod_k f_k$ and $f' = \prod_k f'_k$ such that $\|f_k - f'_k\|_{L_2} \leq \tilde{\epsilon}$ is upper bonded by

$$\|\prod_k f_k - \prod_k f'_k\|_{L_2} = \|\sum_{k=1}^{K} f_1 \ldots f_{k-1}(f_k - f'_k)f'_{k+1} \ldots f'_K\|_{L_2}$$

$$\leq \sum_{k=1}^{K} \|f_1\|_{L_2} \cdots \|f_{k-1}\|_{L_2} \|f_k - f'_k\|_{L_2} \|f'_{k+1}\|_{L_2} \cdots \|f'_K\|_{L_2} \leq K\tilde{\epsilon}.$$

Therefore, if $\{f_{k,j}\}_{j=1}^{N_k}$ be the $\tilde{\epsilon}$-net of $\mathcal{H}_{r,k}(\delta_k)$ where $N_k = \mathcal{N}(\tilde{\epsilon}, \mathcal{H}_{r,k}(\delta_k), L_2(P_\mathcal{X}))$, then the set $\mathcal{E} = \{f = \prod_{k=1}^{K} f_{k,j_k} \mid 1 \leq j_k \leq N_k\}$ is the $K\tilde{\epsilon}$-net of $\mathcal{B}$. Therefore, $\log \mathcal{N}(\tilde{\epsilon}K, \mathcal{B}, L_2(P_\mathcal{X})) \leq \log(\prod_{k=1}^{K} N_k)$. By the entropy condition of $\mathcal{H}_{r,k}$, there exists $c'$ (depending on $c$ and $s$) such that $N_k \leq c'\tilde{\epsilon}^{-2s}$, thus for $\epsilon = \tilde{\epsilon}K$, we have that $\log \mathcal{N}(\epsilon, \mathcal{B}, L_2(P_\mathcal{X})) \leq \sum_{k=1}^{K} c'(\epsilon/K)^{-2s} \leq c'K^{1+2s}\epsilon^{-2s}$. This gives that there exists $C'$ depending on only $s$ and $c$ such that the entropy number of $\mathcal{B}$ is bounded by

$$e_i(\mathcal{B}, L_2(P_\mathcal{X})) \leq C'K^{\frac{1+2s}{2s}} i^{-\frac{1}{2s}}. \tag{S-20}$$

Let $\mathcal{B}(\delta) := \{f = \prod_{k=1}^{K} f_k \mid \|f\|_{L_2} \leq \delta, f_k \in \mathcal{H}_{r,k}, \|f_k\|_{\mathcal{H}_{r,k}} \leq 1\}$ and $\tilde{c}_s = K^{\frac{1+2s}{2s}}$. Then Proposition A.8, the entropy number bound (S-20) and the Infinity-Norm Assumption (Assumption 3) give that

$$\mathrm{E}\left[\sup_{f_k \in \mathcal{H}_{r,k}:\|f_k\|_{\mathcal{H}_{r,k}} \leq 1} \frac{|\frac{1}{n}\sum_{i=1}^{n} \sigma_i \prod_{k=1}^{K} f_k|}{\prod_{k=1}^{K} \|f_k\|_{L_2} + \lambda^{\frac{1}{2}}}\right]$$

$$\leq \mathrm{E}\left[\sup_{f \in \mathcal{B}(\lambda^{1/2})} \frac{|\frac{1}{n}\sum_{i=1}^{n} \sigma_i f(x_i)|}{\|f\|_{L_2} + \lambda^{\frac{1}{2}}}\right]$$

$$\quad + \sum_{i=1}^{\infty} \mathrm{E}\left[\sup_{f \in \mathcal{B}(z^i\lambda^{1/2})\backslash \mathcal{H}_{r,k}(z^{i-1}\lambda^{1/2})} \frac{|\frac{1}{n}\sum_{i=1}^{n} \sigma_i f(x_i)|}{\|f\|_{L_2} + \lambda^{\frac{1}{2}}}\right]$$

$$\leq C'_s \left(\frac{\lambda^{\frac{1-s}{2}}\tilde{c}_s^s}{\lambda^{\frac{1}{2}}\sqrt{n}} \vee \frac{\tilde{c}_s^{\frac{2s}{1+s}}(c_2\lambda^{\frac{1-s_2}{2}})^{\frac{1-s}{1+s}}}{n^{\frac{1}{1+s}}\lambda^{\frac{1}{2}}}\right) + \sum_{i=1}^{\infty} C'_s \left(\frac{z^{i(1-s)}\lambda^{\frac{1-s}{2}}\tilde{c}_s^s}{\sqrt{n}z^{(i-1)}\lambda^{\frac{1}{2}}} \vee \frac{\tilde{c}_s^{\frac{2s}{1+s}}[c_2(z^i\lambda^{\frac{1}{2}})^{1-s_2}]^{\frac{1-s}{1+s}}}{n^{\frac{1}{1+s}}z^{(i-1)}\lambda^{\frac{1}{2}}}\right)$$

$$\leq 4C'_s \left(\frac{1}{1-z^{-s}}\tilde{c}_s^s\sqrt{\frac{\lambda^{-s}}{n}} + \frac{\tilde{c}_s^{\frac{2s}{1+s}}c_2^{\frac{1-s}{1+s}}}{1-z^{-\frac{2s+(1-s)s_2}{1+s}}}\left(\frac{\lambda^{-\frac{1}{2}+\frac{(1-s_2)(1-s)}{2(1+s)}}}{n^{\frac{1}{1+s}}}\right)\right)$$

$$= 4C'_s \left(2\tilde{c}_s^s\sqrt{\frac{\lambda^{-s}}{n}} + \frac{2^{\frac{2s+(1-s)s_2}{s(1+s)}}}{2^{\frac{2s+(1-s)s_2}{s(1+s)}}-1}\tilde{c}_s^{\frac{2s}{1+s}}c_2^{\frac{1-s}{1+s}}\left(\frac{\lambda^{-\frac{2s+(1-s)s_2}{2(1+s)}}}{n^{\frac{1}{1+s}}}\right)\right)$$

$$\leq 4C'_s \left(2 + \frac{2^{\frac{2s+(1-s)s_2}{s(1+s)}}}{2^{\frac{2s+(1-s)s_2}{s(1+s)}}-1}c_2^{\frac{1-s}{1+s}}\right)\left(\tilde{c}_s^s\sqrt{\frac{\lambda^{-s}}{n}} \vee \left(\frac{\tilde{c}_s^{\frac{2s}{1+s}}\lambda^{-\frac{2s+(1-s)s_2}{2(1+s)}}}{n^{\frac{1}{1+s}}}\right)\right).$$

By setting $C_s \leftarrow 4C'_s \left(2 + \frac{2^{\frac{2s+(1-s)s_2}{s(1+s)}}}{2^{\frac{2s+(1-s)s_2}{s(1+s)}}-1}c_2^{\frac{1-s}{1+s}}\right)$, we obtain the assertion. $\square$

The Lemma A.9 gives the following bound.

**Lemma A.10.** *Under the Complexity Assumption (Assumption 2) and the Infinity-Norm Assumption (Assumption 3), there exists a constant $C_s$ depending only on $s, s_2$ and $c, c_2$ such that for all $\lambda > 0$*

$$\mathrm{E}\left[\sup_{f_{(r,k)} \in \mathcal{H}_{r,k}:\|f_{(r,k)}\|_{\mathcal{H}_{r,k}} \leq 1} \frac{|\frac{1}{n}\sum_{i=1}^{n} \epsilon_i \prod_{k=1}^{K} f_{(r,k)}(x_i)|}{\prod_{k=1}^{K} \|f_{(r,k)}\|_{L_2} + \lambda^{\frac{1}{2}}}\right] \leq C_s L \left(\frac{K^{\frac{1+2s}{2}}\lambda^{-\frac{s}{2}}}{\sqrt{n}} \vee \frac{K^{\frac{1+2s}{1+s}}\lambda^{-\frac{2s+(1-s)s_2}{2(1+s)}}}{n^{\frac{1}{1+s}}}\right).$$

*Proof.* By applying the contraction inequality [2, Theorem 4.12] to the bound of Lemma A.9, the assertion is proven. $\qquad\square$

Let $\mathcal{T}_r := \{f - g \mid f = \prod_{k=1}^K f_k, \ g = \prod_{k=1}^K g_k \text{ where } f_k, g_k \in \mathcal{H}_{r,k} \ (k = 1, \ldots, K)\}$. Similarly to Lemma A.9, we have the following bound.

**Lemma A.11.** *Under the Complexity Assumption (Assumption 2) and the Infinity-Norm Assumption (Assumption 3), there exists a constant $\tilde{C}_s$ depending only on $s, s_2$ and $c, c_2$ such that for all $\lambda > 0$*

$$
\mathrm{E}\left[\sup_{(f,f')\in\mathcal{T}_r\times\mathcal{T}_{r'}} \frac{|\frac{1}{n}\sum_{i=1}^n \sigma_i f(x_i) f'(x_i)|}{\|ff'\|_{L_2} + \lambda^{\frac{1}{2}}}\right] \le \tilde{C}_s \left(\frac{K^{\frac{1+2s}{2}}\lambda^{-\frac{s}{2}}}{\sqrt{n}} \vee \frac{K^{\frac{1+2s}{1+s}}}{\lambda^{\frac{2s+(1-s)s_2}{2(1+s)}} n^{\frac{1}{1+s}}}\right).
$$

*Proof.* Let $\mathcal{B} = \{f(x)f'(x) \mid f \in \mathcal{T}_r, \ f' \in \mathcal{T}_{r'}\}$. Along with the same argument with the proof of Lemma A.9, the entropy number of $\mathcal{B}$ is bounded by

$$
e_i(\mathcal{B}, L_2(P_{\mathcal{X}})) \le \tilde{C}' K^{\frac{1+2s}{2s}} i^{-\frac{1}{2s}},
$$

where $\tilde{C}'_s$ is a constant depending on only $s$ and $c$. Then, using the pealing device as in Lemma A.9, we obtain the assertion. $\qquad\square$

Let the upper bound given in Lemmas A.9 and A.11 be $\zeta_n(\lambda)$:

$$
\zeta_n(\lambda) = \zeta_n := \max\{C_s, \tilde{C}_s\} \left(\frac{K^{\frac{1+2s}{2}}\lambda^{-\frac{s}{2}}}{\sqrt{n}} \vee \frac{K^{\frac{1+2s}{1+s}}}{\lambda^{\frac{2s+(1-s)s_2}{2(1+s)}} n^{\frac{1}{1+s}}}\right),
$$

where $C_s$ and $\tilde{C}_s$ are the constants appeared in each lemma respectively. Lemma A.9 and Lemma A.11.

In addition to Lemma A.11, we obtain the following tail probability bound.

**Lemma A.12.** *Under the Complexity Assumption (Assumption 2) and the Infinity-Norm Assumption (Assumption 3), there exists a universal constant $C > 0$ such that, for any $0 < \lambda$, it holds that*

$$
\sup_{f\in\mathcal{T}_r, f'\in\mathcal{T}_{r'}} \left|(P - P_n)\left(\frac{ff'}{\|ff'\|_{L_2} + \lambda^{\frac{1}{2}}}\right)\right| \le C\zeta_n \max\{1, \tau\}
$$

*with probability $1 - \exp(-\tau)$ for all $\tau > 0$.*

*Proof.* We apply Talagrand's concentration inequality [6, 1]. To apply Talagrand's inequality, we need to bound the $L_2$-norm and the sup-norm of each term in the supremum in the LHS. They are bounded as

$$
\mathrm{E}_X\left(\frac{(f(X)f'(X))^2}{(\|ff'\|_{L_2} + \lambda^{\frac{1}{2}})^2}\right) \le 1, \quad \frac{|f'(X)f(X)|}{\|ff'\|_{L_2} + \lambda^{\frac{1}{2}}} \le \frac{4}{\lambda^{1/2}}.
$$

Moreover, by Eq. (S-19), it holds that

$$
\mathrm{E}[\sup_{(f,f')\in\mathcal{T}_r\times\mathcal{T}_{r'}} |(P - P_n)(ff'/(\|ff'\|_{L_2} + \lambda^{1/2}))|]
$$

$$
\le 2\mathrm{E}\left[\sup_{(f,f')\in\mathcal{T}_r\times\mathcal{T}_{r'}} \left|\frac{1}{n}\sum_{i=1}^n \sigma_i(f(x_i)f'(x_i)/(\|ff'\|_{L_2} + \lambda^{1/2}))\right|\right].
$$

Therefore, by Talagrand's concentration inequality and Lemma A.10, there exists a universal constant $C > 0$ such that

$$
P\left(\sup_{f,f'\in\mathcal{T}_r\times\mathcal{T}_{r'}} \frac{|\frac{1}{n}\sum_{i=1}^n \sigma_i f(x_i) f'(x_i)|}{\|ff'\|_{L_2} + \lambda^{\frac{1}{2}}} \ge C\left[2\zeta_n + \sqrt{\frac{\tau}{n}} + \frac{4\lambda^{-1/2}\tau}{n}\right]\right) \le e^{-\tau},
$$

for all $\tau > 0$. By the definition of $\zeta_n$, the right hand side is upper bounded by $7C\zeta_n \max\{1, \tau\}$. Then, we obtain the assertion. $\qquad\square$

Using the same argument, the following bound also holds.

**Corollary A.13.** *Under the Complexity Assumption (Assumption 2) and the Infinity-Norm Assumption (Assumption 3), there exists a universal constant $\tilde{C} > 0$ such that, for any $0 < \lambda$, it holds that*

$$\max_{1 \leq r, r' \leq d} \sup_{f \in \mathcal{T}_r, f' \in \mathcal{T}_{r'}} \left| (P - P_n) \left( \frac{f f'}{\|f f'\|_{L_2} + \lambda^{\frac{1}{2}}} \right) \right| \leq \tilde{C} \log(d) \zeta_n \max\{1, \tau\}$$

*with probability $1 - \exp(-\tau)$ for all $\tau > 0$.*

*Proof.* Taking the uniform bound with respect to $r, r'$ of Lemma A.13. We obtain the assertion. $\square$

Let $\tilde{\mathcal{T}}_{r,k} = \{(f_{(r,k)} - f'_{(r,k)})(\prod_{k' \neq k} f_{(r,k')} - \prod_{k' \neq k} f'_{(r,k')}) \mid f_{(r,k')} \in \mathcal{H}_{r,k}, f'_{(r,k')} \in \mathcal{H}_{r,k'} \ (k' = 1, \ldots, K)\}$. Then by the same argument as Corollary A.13, we have the following lemma.

**Lemma A.14.** *Under the Complexity Assumption (Assumption 2) and the Infinity-Norm Assumption (Assumption 3), there exists a universal constant $\tilde{C}' > 0$ such that, for any $0 < \lambda$, it holds that*

$$\max_{1 \leq r \leq d, 1 \leq k \leq K} \sup_{f \in \tilde{\mathcal{T}}_{r,k}} \left| \frac{1}{n} \sum_{i=1}^{n} \left( \frac{\epsilon_i f(x_i)}{\|f\|_{L_2} + \lambda^{\frac{1}{2}}} \right) \right| \leq \tilde{C}' L \log(dK) \zeta_n \max\{1, \tau\},$$

$$\max_{1 \leq r \leq d, 1 \leq k \leq K} \sup_{f, f' \in \tilde{\mathcal{T}}_{r,k}} \left| (P - P_n) \left( \frac{f f'}{\|f f'\|_{L_2} + \lambda^{\frac{1}{2}}} \right) \right| \leq \tilde{C}' \log(dK) \zeta_n \max\{1, \tau\}$$

*with probability $1 - \exp(-\tau)$.*

The proof is almost identical to that of Corollary A.13.

Let $\mathcal{T}'_{r,k} = \{(f_{(r,k)}(x) - f'_{(r,k)}(x)) \prod_{k' \neq k}^{K} f^{**}_{(r,k')}(x) \mid f_{(r,k)}, f'_{(r,k)} \in \mathcal{H}_{r,k}, \|f_{(r,k)}\|_{\mathcal{H}_{r,k}} \leq 1, \|f'_{(r,k)}\|_{\mathcal{H}_{r,k}} \leq 1\}$. Then Lemma A.9 gives the following bound.

**Lemma A.15.** *Under the Complexity Assumption (Assumption 2) and the Infinity-Norm Assumption (Assumption 3), there exists a constant $C'_s$ depending only on $s$ and $c$ such that for all $\lambda > 0$*

$$\mathrm{E} \left[ \sup_{f \in \mathcal{T}'_{r,k}} \frac{|\frac{1}{n} \sum_{i=1}^{n} \sigma_i f(x_i)|}{\|f\|_{L_2} + \lambda^{\frac{1}{2}}} \right] \leq C'_s \left( \frac{\lambda^{-\frac{s}{2}}}{\sqrt{n}} \vee \frac{1}{\lambda^{\frac{1}{2}} n^{\frac{1}{1+s}}} \right).$$

Let

$$\zeta'_n = C'_s \left( \frac{\lambda^{-\frac{s}{2}}}{\sqrt{n}} \vee \frac{1}{\lambda^{\frac{1}{2}} n^{\frac{1}{1+s}}} \right) \tag{S-21}$$

where $C'_s$ is given in Lemma A.15. Note that $\zeta'_n$ is independent of $K$ while $\zeta_n$ depends on it. Then, going through the same argument as Lemmas A.10, A.12 and A.13, we obtain the following lemma.

**Lemma A.16.** *Under the Complexity Assumption (Assumption 2) and the Infinity-Norm Assumption (Assumption 3), there exists a universal constant $C'$ such that all of the following three inequalities are satisfied more than probability $1 - \exp(-\tau)$ for all $\tau > 0$:*

$$\max_{1 \leq r \leq d, 1 \leq k \leq K} \sup_{f \in \mathcal{T}'_{r,k}} \left| (P - P_n) \left( \frac{f^2}{\|f\|_{L_2} + \lambda^{\frac{1}{2}}} \right) \right| \leq C' \log(dK) \zeta'_n \max\{1, \tau\},$$

$$\max_{1 \leq r \leq d, 1 \leq k \leq K} \sup_{f \in \mathcal{T}'_{r,k}} \left| \frac{1}{n} \sum_{i=1}^{n} \left( \frac{\epsilon_i f(x_i)}{\|f\|_{L_2} + \lambda^{\frac{1}{2}}} \right) \right| \leq C' L \log(dK) \zeta'_n \max\{1, \tau\},$$

$$\max_{1 \leq r \leq d, 1 \leq k \leq K} \sup_{f \in \mathcal{H}_{r,k}, \|f\|_{\mathcal{H}_{r,k}} \leq 1} \left| (P - P_n) \left( \frac{f^2}{\|f\|_{L_2} + \lambda^{\frac{1}{2}}} \right) \right| \leq C' \log(dK) \zeta'_n \max\{1, \tau\}.$$