[Reviews · NeurIPS 2016]

Reviewer 1

Summary

This paper analyzes a coordinate descent style algorithm for low-rank tensor regression with factors in RKHS. The main result is that after sufficient rounds of the algorithm, the error rates of the estimator is minimax optimal up to a log-factor (assuming the initialization is not far away).

Qualitative Assessment

This paper analyzes a coordinate descent style algorithm for low-rank tensor regression with factors in RKHS. The main result is that after sufficient rounds of the algorithm, the error rates of the estimator is minimax optimal up to a log-factor. The results hold only if the estimator is initialized at a sufficiently close neighborhood of the true function. The assumptions made are pretty much standard in RKHS literature. The paper is technically strong and the investigation is thorough. Experimental results are provided on different data sets that validate the theory. I went over the proof in some detail (although not so throughly) and it seems to me there are no significant issues. The main drawbacks of the paper are: (i) the assumption that P(X) is a product measure - this is a significant assumption and should be mentioned up front in the introduction section itself and (2) presence of the log(dK) term in the rates which could probably be removed with a more refined analysis. Furthermore, a more realistic analysis would be to assume the true function is not in RKHS and to expresses the error as sum of estimation and approximation error - which make more challenging the issue of initializing the estimator (and also adaptation properties (with respect to 'd') come into play).

Confidence in this Review

2-Confident (read it all; understood it all reasonably well)


Reviewer 2

Summary

This paper presents a new non-parametric tensor regression method based on kernels. More specifically, the authors proposed a regularization based optimization approach with alternating minimization for non-parametric tensor regression model [15]. Moreover, the theoretical guarantee for the proposed method is presented the paper. Through experiments on various datasets, the proposed method compares favorably with existing state-of-the-art.

Qualitative Assessment

This paper presents a new non-parametric tensor regression method based on kernels. More specifically, the authors proposed a regularization based optimization approach with alternating minimization for non-parametric tensor regression model [15]. Moreover, the theoretical guarantee for the proposed method is presented the paper. Through experiments on various datasets, the proposed method compares favorably with existing state-of-the-art. The paper is clearly written and easy to read. I understand the key contribution of this paper is the theoretical analysis of the non-parametric tensor regression. However, the experimental section of this paper can be improved (See comments below). Overall, the proposed method is simple yet effective and theoretically guaranteed. I am happy to see the paper in NIPS. Detailed comments: 1. In experiments, the performance of linear and nonlinear methods becomes similar when the number of samples increases. Are there any gap between linear and nonlinear methods with bigger sample size (e.g., 5000)? Also, including more comparison to other state-of-the-art linear tensor regression methods would be useful. 2. Related to the first question. If the performance of the linear and nonlinear methods are similar when the number of samples are large. Then, the experiments with web related data is not so practically interesting (though it is worth validating the method), since the web-data tends to be hundred millions of data points. That is, we might need to simply use a linear method for web-related datasets. Thus, if authors can include some experiments with high-dimensional and small number of samples datasets such as bioinformatics related datasets, the proposed method is more practically interesting. 3. How scale the AMP method with respect to the number of dimension? The method still works for d = 10^4 or more? 4. How the performance of the scaled latent convex regularization method? For completeness, it would be great to include it in either main text or supplementary materials. 5. How fast the AMP algorithm? There is no time comparison in Fig 1. (Author claimed the AMP method is faster in the main text without time comparison). 6. How to initialize the proposed method?

Confidence in this Review

2-Confident (read it all; understood it all reasonably well)


Reviewer 3

Summary

The paper studies the problem of kernel nonparametric tensor learning proposed in previous work. In this problem of tensor learning, each data points has k blocks of coordiantes, and the ground truth function is a sum of d sub-fucntions, where the r-th sub-fucntions is a product of K component functions f_(r,k) and f_(r,k) is applied on the k-th block of coordiantes of the data point. In kernel nonparametric setting, each component function is assumed to be from some RKHS H_{r,k}. The alternating minimization method is that in each round, one update one f_{r,k} while fixing the others. The paper shows that when the initialization is close enough to the ground truth, the alternating minimization has a linear convergence rate, and the generalization error achieves the minimax optimality.

Qualitative Assessment

For linear tensor learning, bounds for both Bayesian/convexified algorithms and alternating minimization are known. For nonparametric tensor learning, only the bound for Bayesian/convexified algorithms is known in previous work. Since alternating minimization has good practical performance and is more computationally effficient, it is interesting to prove the bounds for it. pros: 1. The bound for alternating minimization has linear rate in the sense that the error has two terms: the first corresponds to the generalization error while the second corresponds to the optimization error which decreases exponentiall with the number of epochs in the algorithm. 2. The generalization error term is minimax optimal (matching the lower bound in the previous work). cons: 1. The initialization requirement in Thm 3 is quite strong: the init is already very close to the ground truth. It will be more interesting if one can show a milder requirement, eg, the distance is constant instead of v_min.(The authors seem to view v_min as constants, but in practice this v_min can be quite small so getting rid of v_min or better dependence on v_min will be better) 2. Design algorithms to get the needed initialization. The authors mentioned that one can used the Bayesian estimator in previous work to do that but if the initialization requirement is strong, this loses the computational benefits.

Confidence in this Review

2-Confident (read it all; understood it all reasonably well)


Reviewer 4

Summary

This paper studies an alternating regularized least squares algorithm for solving tensor-based kernel ridge regression problem. The main results obtained in this paper are two-fold: First, under some boundedness assumptions and the so-called "incoherence assumption", the generalization bound of the solution at each iteration is derived. Second, the linear convergence rate of the above algorithm can be obtained under certain assumptions.

Qualitative Assessment

The paper is well-polished, presented neatly, and is easy to follow. The theoretical analysis conducted here seems sound. My only concern in this paper is the assumption on the incoherence (Definition 1 and Assumption 4). Clearly, it is an extension and generalization of the incoherence assumption for linear models. However, noticing that the considered regression problem learns functions from a reproducing kernel Hilbert space, the incoherence assumption essentially assumes the incoherence property of the Gram matrix. This, in my view, could be quite stringent (taking the RBF kernel for instance). Therefore, I would suggest that the authors should give some more comments or even numerical illustrations towards this assumption.

Confidence in this Review

2-Confident (read it all; understood it all reasonably well)


Reviewer 5

Summary

This paper theoretically analyzes the alternating minimization procedure for the nonparametric tensor regression model. It is shown that the alternating minimization algorithm converges to an optimal solution in linear rate, and the generalization error of the estimator achieves the minimax optimality (up to a log scale) with proper assumption on the initialization. Overall, this paper is well written and it addresses an interesting and difficult problem. This paper is suitable for NIPS due to its theoretical contributions in nonlinear tensor learning.

Qualitative Assessment

I have a few comments regarding the current paper. Q1: Initialization: Theoretically, the initialization is only required to be of a constant distance to the truth. How to gurantee this in practice? Which initial values are used in the experiments? Q2: Computational complexity analysis: As claimed in the paper, an advantage of the alternating minimization procedure is its faster computation than the Bayesian method GP-MTL. Can the authors provide detailed computational complexity analysis of the proposed method and report the computational time of both methods in the experiments? Q3: Experiments: In literature, there are many kernel-based multi-task learning methods. In the experiments, the authors only compared their method with another nonparametric tensor method GP-MTL. It is of interest to know its numeric comparison with state-of-art nonlinear multi-task learning methods. Q4: Literature review: there are a few other related literature on provable tensor learning via alternative minimization methods, see for example the following papers. The authors are encouraged to comment the connection of the proposed method with these papers. (1) Anandkumar, A., Ge, R. and Janzamin, M. (2014), Guaranteed non-orthogonal tensor decomposition via alternating rank-1 updates. (2) Sun, W., Lu, J., Liu, H., and Cheng, G. (2016), Provable sparse tensor decomposition.

Confidence in this Review

3-Expert (read the paper in detail, know the area, quite certain of my opinion)


Reviewer 6

Summary

In this paper the authors study an alternating minimization procedure for nonparametric tensor learning. The authors proved that the alternating estimation procedure yields an estimator with a minimax-optimal statistical rate of convergence with a logarithmic number of iterations given a sufficiently accurate initialization.

Qualitative Assessment

In this paper the authors study an alternating minimization procedure for nonparametric tensor learning. The authors proved that the alternating estimation procedure yields an estimator with a minimax-optimal statistical rate of convergence with a logarithmic number of iterations given a sufficiently accurate initialization. This result significantly adds to the recent line of work on alternating minimization on non-convex objective functions by incorporating nonparametric analysis for the first time. This paper is well written and the proof is technically sound. The only minor concern I have is about the requirement on the initialization. In nonparametric settings, the O(1) accuracy requirement on the initialization is quite stringent. Although the authors pointed out this can be achieved by Bayesian procedures, I did not see why this is feasible intuitively. It would be better if the authors can shed more light upon this issue with some discussion in the paper.

Confidence in this Review

3-Expert (read the paper in detail, know the area, quite certain of my opinion)